# Over-Alignment vs Over-Fitting:
# The Role of Feature Learning Strength in Generalization

**Taesun Yeom** [1]   **Taehyeok Ha** [1]   **Jaeho Lee** [1]

## Abstract

Feature learning strength (FLS), i.e., the inverse of the effective output scaling of a model, plays a critical role in shaping the optimization dynamics of neural nets. While its impact has been extensively studied under the asymptotic regimes—both in training time and FLS—existing theory offers limited insight into how FLS affects generalization in practical settings, such as when training is stopped upon reaching a target training risk. In this work, we investigate the impact of FLS on generalization in deep networks under such practical conditions. Through empirical studies, we first uncover the emergence of an *optimal FLS*—neither too small nor too large—that yields substantial generalization gains. This finding runs counter to the prevailing intuition that stronger feature learning universally improves generalization. To explain this phenomenon, we develop a theoretical analysis of gradient flow dynamics in two-layer ReLU nets trained with logistic loss, where FLS is controlled via initialization scale. Our main theoretical result establishes the existence of an optimal FLS arising from a trade-off between two competing effects: An excessively large FLS induces an *over-alignment* phenomenon that degrades generalization, while an overly small FLS leads to *over-fitting*.

## 1. Introduction

One of the key mysteries of deep learning is its ability to find well-generalizing solutions, even when severely over-parametrized (Zhang et al., 2017). Because this behavior appears to contradict classical learning theory, a number of explanations have been proposed. A leading hypothesis is based on *implicit bias*—the tendency of neural nets to favor learning certain solutions, even in the absence of an explicit regularization (Vardi, 2023). A growing body of work investigates the origins of this phenomenon, attributing it to various factors such as gradient-based optimization dynamics (Soudry et al., 2018; Lyu & Li, 2020), model architecture (Teney et al., 2024; Cao et al., 2023), or hyperparameters, e.g., the learning rate (Even et al., 2023; Wu et al., 2023).

Among many factors, the *feature learning strength* (FLS) stands out as particularly important (Woodworth et al., 2020; Atanasov et al., 2025). FLS is defined as the inverse of the effective scaling applied to the model output, which is typically controlled by the initialization scale or an explicit output multiplier, such as the softmax temperature. Varying FLS leads to two qualitatively distinct training regimes. When FLS is large, features evolve nonlinearly throughout training, reflecting genuine feature learning (Woodworth et al., 2020; Atanasov et al., 2022). In contrast, when FLS is small, training closely resembles kernel learning, with features remaining largely fixed (Jacot et al., 2018; Chizat et al., 2019). A substantial body of prior works has shown that analyzing these two regimes yields valuable insights into the optimization dynamics and generalization of deep learning (Arora et al., 2019b; Allen-Zhu et al., 2019; Sclocchi et al., 2023; Atanasov et al., 2025; Dominé et al., 2025; Simon et al., 2026).

However, our theoretical understanding of how FLS affects generalization remains poorly aligned with practical observations. This gap is twofold. First, existing theories offer little concrete guidance for tuning FLS-related hyperparameters to achieve optimal generalization. Their conclusions often reduce to the coarse message that "stronger feature learning improves generalization" (Woodworth et al., 2020; Atanasov et al., 2025), whereas in practice, intermediate levels of feature learning—neither too weak nor too strong—tend to perform best (Agarwala et al., 2023; Masarczyk et al., 2025). Second, much of the theoretical literature focuses on properties of the limiting solution, which is rarely relevant in real training settings. In practice, training is typically halted once a target training risk is reached or a fixed optimization budget is exhausted. Since stronger feature learning generally requires more optimization steps, conclusions drawn

---

[1]Pohang University of Science and Technology (POSTECH), Pohang, South Korea. Correspondence to: Jaeho Lee <jaeho.lee@postech.ac.kr>.

*Proceedings of the 43rd International Conference on Machine Learning*, Seoul, South Korea. PMLR 306, 2026. Copyright 2026 by the author(s).

from the limiting regime can be misleading when applied to finite-time training (Woodworth et al., 2020).

**Contribution.** In this work, we aim to narrow the gap between the FLS-based theoretical understanding and practice, by studying the following two research questions:

> - **Q1.** Does stronger feature learning always help generalization, under practical setups?
> - **Q2.** If not, can we explain such a gap theoretically?

To address **Q1**, we conduct experiments on image classification tasks using VGG (Simonyan & Zisserman, 2015) and ResNet (He et al., 2016) architectures. We find that, even when models achieve perfect training accuracy or attain the same training risk, their generalizability differs significantly depending on the FLS. Surprisingly, across all datasets and architectures we consider, excessively large FLS values consistently harm generalization, and an intermediate optimal FLS emerges, in contrast with the prevailing belief (Figure 1). Moreover, we find that the benefit of tuning FLS grows with task complexity: as the dataset's intrinsic dimensionality increases, selecting the optimal FLS yields increasingly large generalization gains.

Motivated by these empirical results, we proceed to address **Q2** by analyzing the optimization dynamics induced by varying the FLS. Building on recent work that studies gradient flow dynamics in the strong feature learning regime (Min et al., 2024; Boursier & Flammarion, 2025), we first establish that FLS—equivalently, the initialization scale in our setting—critically governs the angular deviation of the weights (or the induced predictor) throughout training (Lemmas 5.2 and 5.4). Leveraging this characterization, we derive an error bound for binary Gaussian mixtures and decompose it into two distinct components: a data-dependent *over-alignment* term and an *over-fitting* term (Theorem 5.6). This decomposition exposes a fundamental trade-off between strong and weak feature learning and implies the existence of a data-dependent optimal FLS. Together, these results capture what is observed in practice, providing a fresh perspective on how FLS shapes generalization.

In summary, our work provides both empirical and theoretical results providing insights into practical implicit bias in classification tasks, which has been largely unexplored in prior studies. In particular, we emphasize the role of feature learning strength in shaping generalization behavior in deep learning. We hope our work serves as a step toward demystifying the generalization ability of neural networks.

**Notation.** Scalars, vectors, and matrices are denoted by lowercase (e.g., $a$), bold lowercase (e.g., $\mathbf{a}$), and bold uppercase letters (e.g., $\mathbf{A}$), respectively. The norm $\|\cdot\|$ denotes the Euclidean norm for vectors, and the spectral norm for matrices. $\|\cdot\|_F$ denotes the Frobenius norm. $\mathbf{I}_n$ denotes the $n \times n$ iden-

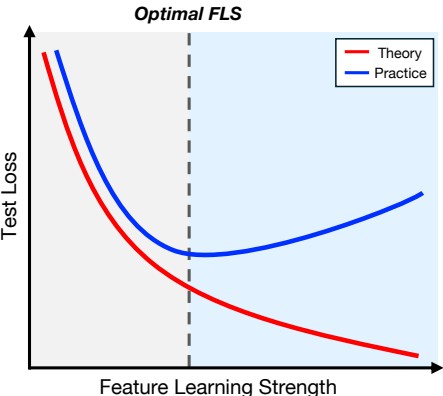

*Optimal FLS*

*Figure 1.* **Emergence of an optimal FLS.** We empirically observe that, under standard classification setups, stronger feature learning tends to degrade generalization performance of the model when it exceeds a certain threshold, implying the existence of an "optimal FLS" that is neither too large nor too small.

tity matrix, $\angle(\cdot, \cdot)$ denotes the angle between two vectors, $\Phi(\cdot)$ is the Gaussian CDF, and $\mathbb{R}_+ := \{x \in \mathbb{R} : x > 0\}$.

## 2. Related Work

**FLS in deep learning.** The optimization behavior of neural nets is highly sensitive to the *feature learning strength* (FLS), i.e., the inverse of the effective scaling of the model output (Chizat et al., 2019; Atanasov et al., 2025). This scaling can be controlled through the weight initialization scheme (Woodworth et al., 2020; Kunin et al., 2024; Yeom et al., 2025) or by explicitly rescaling the model output (Chizat et al., 2019; Atanasov et al., 2025). Varying this scale induces a transition between two distinct regimes: a feature learning regime at small scales, and kernel regime at large scales. In the strong feature learning regime, the training dynamics are highly nonlinear, inducing phenomena such as neuron alignment (Maennel et al., 2018; Min et al., 2024; Boursier & Flammarion, 2025) or saddle-to-saddle dynamics (Jacot et al., 2021; Kunin et al., 2025). In contrast, in the kernel regime, the network behaves approximately linearly with respect to its initialization, with little updates in features (Jacot et al., 2018; Chizat et al., 2019). Majority of these works aim to provide a clear picture of learning dynamics itself, induced by the gradient descent. Our work, on the other hand, focuses on the generalization performance of the models induced by these training dynamics.

**Feature learning and generalization.** A widely held belief is that the stronger feature learning always leads to better generalization in standard—i.e., in-distribution—classification. In such regime, the training dynamics result in sparse features, which in turn leads to a better generalization (Woodworth et al., 2020; Li et al., 2021; Stöger & Soltanolkotabi, 2021; Li et al., 2023). Several prior works

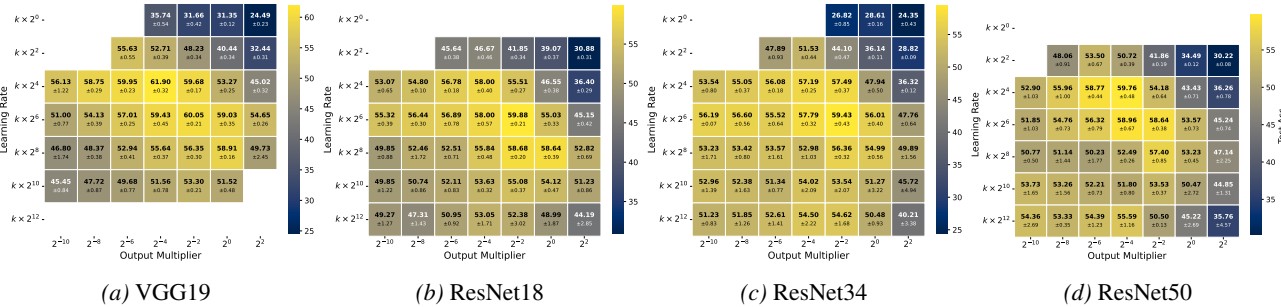

*Figure 2.* **Emergence of optimal FLS in generalization.** Peak test accuracy (%) of various networks trained on CIFAR-100. Blank grids indicate cases where, for at least one of the three seeds, the training accuracy does not exceed 99%. For readability, the learning rate axis is labeled using the pre-normalized values (i.e., $\eta$), where $k = 6.4 \times 10^{-4}$. Further details can be found in Section A.1.

establish concrete connections to the generalization: Sclocchi et al. (2023) study the phase diagram varying SGD noise and feature learning strength; in an online learning setup, Atanasov et al. (2025) empirically analyze the generalization behavior across varying feature learning strength and learning rate; most similar to our work, Petrini et al. (2022) study generalization behavior in a spherical regression task. This work, however, mainly considers the two extreme choices of FLS in infinite-width networks: mean-field (Mei et al., 2018) vs. neural tangent kernel. In contrast, our work primarily focuses on characterizing the *optimal* feature learning strength which lies between these regimes, in the classification setup.

## 3. Empirical Takes on FLS & Generalization

In this section, we empirically study how FLS affects generalization in deep networks and present a nontrivial observation that has not been discussed in prior literature: *Larger FLS can hurt generalization in standard classification.*

**Controlling FLS in deep networks.** To control FLS, in this section we consider the following scaling rule for neural networks. Suppose we train a neural network $f$ via gradient descent, with learning rate $\eta$. Here, we rescale the function as $f \mapsto cf$ for some output multiplier $c > 0$ and set the learning rate to $\eta/c$. Here, a smaller $c$ corresponds to a larger FLS. This scaling scheme is analogous to the way FLS is controlled in a widely used parameterization scheme, the so-called *maximal update parameterization* (Geiger et al., 2020; Yang & Hu, 2021; Bordelon & Pehlevan, 2022) and temperature scaling (Agarwala et al., 2023; Masarczyk et al., 2025); see Section B.1 for more details.

### 3.1. Emergence of the sweet spot in generalization

In previous empirical work, FLS is regarded as a trade-off quantity between computational resources and generalization (Woodworth et al., 2020); that is, larger FLS, which typically requires longer training time, leads to better generalization. To check whether this holds for deep networks, we conduct standard image classification experiments using

widely used architectures: VGG19 with batch normalization, and ResNet{18, 34, 50}. We use CIFAR-10 and CIFAR-100 as representative datasets for the classification task. Here, we present the results of the CIFAR-100; see Section A.4 for additional experimental results, including CIFAR-10.

Since the FLS parameter $c$ is heavily influenced by the choice of the learning rate $\eta$ (Atanasov et al., 2025), we sweep over different values of $c$ and $\eta$ and present test-accuracy heatmaps on the $(c, \eta/c)$ plane in Figure 2. Each grid point is averaged over three random seeds, and we report the mean with its standard deviation. For a fair comparison, all networks are trained until they achieve near-perfect training accuracy (i.e., above 99 percent).

Taking a closer look at Figure 2, we observe that, as reported in prior work, training with a larger $c$ tends to degrade generalization (Woodworth et al., 2020; Mehta et al., 2021). On the other hand, the interesting observation here is that there exists an *optimal FLS*: output multipliers below the optimum also hurt generalization, and this trend holds across all networks. Notably, these results have been obtained when the networks have already reached their peak generalization performance (i.e., further training leads to overfitting rather than improvement). Therefore, our findings directly refute the common claim that "with sufficiently long training, a larger FLS is better in classification."

Moreover, our empirical results reveal the practical benefits of using an optimal FLS. In typical settings, hyperparameter tuning does not explicitly include the FLS. However, in such cases—for example, when only the learning rate is tuned— one may fail to reach best generalization, even if the model appears well-optimized within the chosen search space. For instance, in Figure 2d, the best test accuracy achieved with the default scale (i.e., $c = 2^0$) is surpassed by that achieved with the optimal FLS (i.e., $c = 2^{-4}$) by about 6% (e.g., 53.57% vs. 59.76%). These observations suggest that FLS should be treated as a critical axis for hyperparameter tuning, alongside conventional choices.

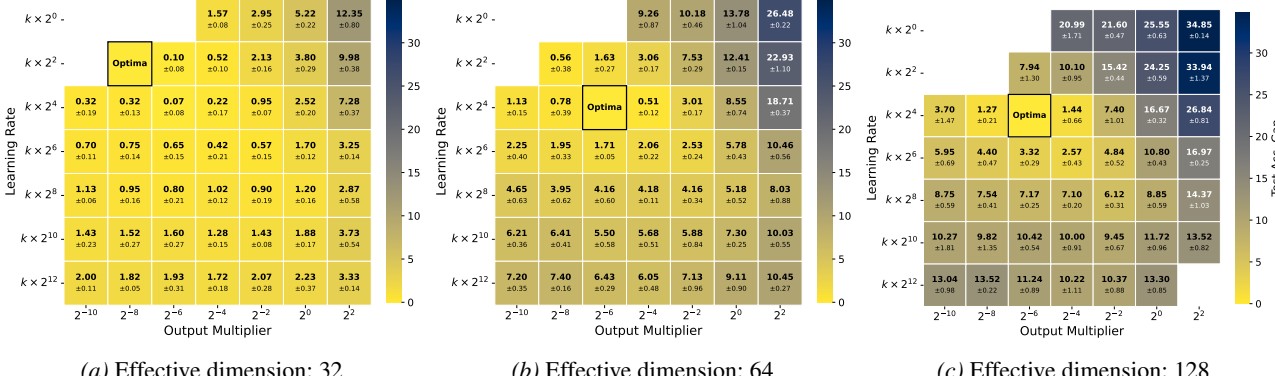

*(a)* Effective dimension: 32   *(b)* Effective dimension: 64   *(c)* Effective dimension: 128

*Figure 3.* **Optimal FLS is more beneficial for the difficult dataset.** The gap of the peak test accuracy (%) of ResNet18 trained on a BigGAN-generated dataset against the best FLS. We have varied the effective dimensionality of the samples generated, to control the task difficulty. Blank grids indicate the cases where, for at least one of the three seeds, the training accuracy does not exceed 99%. For readability, the learning rate axis is labeled using the pre-normalized values (i.e., $\eta$), where $k = 6.4 \times 10^{-4}$.

### 3.2. Benefits of optimal FLS and the task difficulty

Having observed an optimal FLS in standard classification tasks, we now examine how this behavior changes under varying conditions, such as datasets with different levels of difficulty. In this subsection, we focus on the relationship between FLS and the *intrinsic dimensionality* of a dataset, a notion for characterizing its complexity (Ansuini et al., 2019; Gong et al., 2019).

Recent work by Pope et al. (2021) studies the impact of a dataset's intrinsic dimensionality on generalization by explicitly varying the effective dimensionality of the latent vector: Specifically, by zeroing out a predefined subset of indices in each input random vector of deep generative models. They find that higher intrinsic dimensionality increases sample complexity. Following this work, we generate a synthetic dataset with 10 classes from the dog category in ImageNet using pretrained BigGAN (Brock et al., 2019). We vary the effective dimensionality among 32, 64, and 128; for brevity, we refer to each dataset by its effective dimensionality. See Section A.1 for more details.

In Figure 3, we present a heatmap of "the gap in peak test accuracy" for ResNet18 across datasets with different effective dimensionalities. The gap is defined as the difference between the highest accuracy on the heatmap (i.e., at 'Optima' in each heatmap) and the accuracy of each grid cell. Consistent with Figure 2, we again observe a "sweet spot" of generalization, and the network in this regime consistently outperforms other configuration across datasets. Most notably, as the effective dimensionality increases (i.e., as the task becomes more difficult), the benefit of using the optimal FLS becomes larger, as can be seen from the increased gap in the test accuracy.

As a takeaway, these results highlight the practical value of identifying the optimal FLS. The advantage of doing so is particularly more pronounced for challenging tasks.

We refer readers to Section A.5 for additional results, covering different architectures and evaluation metrics.

## 4. Problem Formulation

To demystify the internal mechanisms of the phenomenon observed in Section 3—the emergence of the optimal FLS—we move onto a theoretical analysis. This section describes the problem formulation and relevant preliminaries, based on which we establish the theoretical results in Section 5.

### 4.1. Preliminaries: Two-phase dynamics

First, we describe some known results about the optimization dynamics of the models under various FLS. In particular, we focus on the case of *large FLS*—the models with small feature learning strengths can be understood easily, as they can be approximated by their linearized functionals.

For large FLS, models with positively homogeneous activations (e.g., ReLU) exhibit an interesting learning dynamics. Roughly, their training consists of two distinct phases.

- *Phase 1: Neuron alignment.* In the early phase, weights are aligned to particular direction, with only slight growth in output scale and marginal decrease of the loss. This phenomenon is known as *neuron alignment* (or *directional convergence*) (Maennel et al., 2018; Ji & Telgarsky, 2019).
- *Phase 2: Margin maximization.* After neurons are aligned, the loss begins to decrease more noticeably. As the activation patterns have been stabilized in *Phase 1*, the model in phase 2 behaves approximately as a linear model, whose optimization is well understood (Ji & Telgarsky, 2019; Arora et al., 2019a; Nacson et al., 2019; Lyu & Li, 2020).

In Section 5, we will demonstrate that an analysis on the large FLS regime alone suffices to establish the optimality

of the intermediate FLS, which is neither too large nor too small. Nevertheless, we will also show that our theory can be extended to the case of small FLS, in agreement with the empirical results presented.

## 4.2. Formulation

Now we describe the theoretical setup we consider.

Consider a binary classification task with a $d$-dimensional input $\mathbf{x} \in \mathbb{R}^d$ and a binary label $y \in \{-1, +1\}$. As the classifier, we consider a bias-free two-layer neural network using the ReLU activation:

$$f(\mathbf{x}; \theta) = \sum_{j=1}^{h} v_j \sigma(\langle \mathbf{w}_j, \mathbf{x} \rangle). \tag{1}$$

Here, $\sigma(x) = \max\{0, x\}$ denotes the ReLU activation and $\theta := (\mathbf{W}, \mathbf{v})$ denotes the tuple of parameters, with the first layer parameters $\mathbf{W} = [\mathbf{w}_1, \cdots, \mathbf{w}_h] \in \mathbb{R}^{d \times h}$ and the second layer parameter $\mathbf{v} = [v_1, \cdots, v_h]^\top \in \mathbb{R}^h$.

The training dataset consists of $n$ independently drawn samples $D = \{(\mathbf{x}_i, y_i)\}_{i=1}^n$ and define $\mathbf{x}_{\max} := \max_i \|\mathbf{x}_i\|$. Using this dataset, we minimize the *training risk* over the samples, defined as the following.

$$\hat{L}(\theta) := \frac{1}{n} \sum_{i=1}^{n} \ell(f(\mathbf{x}_i; \theta), y_i). \tag{2}$$

We use the logistic loss, i.e., $\ell(\hat{y}, y) = \log(1 + \exp(-y\hat{y}))$, where $\hat{y}$ denotes the model output.[1] The training risk is minimized via gradient flow. More concretely, we conduct

$$d\mathbf{W}/dt \in -\partial_{\mathbf{W}} \hat{L}(\theta), \qquad d\mathbf{v}/dt \in -\partial_{\mathbf{v}} \hat{L}(\theta), \tag{3}$$

where $\partial$ denotes the Clarke subdifferential (Clarke, 1975).

**Feature learning strength.** The FLS is controlled with the *scale factor* $\alpha > 0$ of the initialization. Precisely, the first layer weight $\mathbf{W}$ is initialized in two steps: First, we sample the entries of the reference matrix $\mathsf{W}$ from some distribution $\mathcal{P}$. Then, we scale this weight by $\alpha$ to initialize $\mathbf{W}$, i.e.,

$$\mathbf{W}(0) = \alpha \mathsf{W}. \tag{4}$$

Here, we define the quantity $\mathsf{W}_{\max} := \max_{j \in [h]} \|\mathsf{W}_j\|$. The second layer weights are determined as

$$v_j(0) \sim \mathrm{Unif}(\{\|\mathbf{w}_j(0)\|, -\|\mathbf{w}_j(0)\|\}) \tag{5}$$

This initialization scheme enables us to utilize existing tools for gradient flow analysis, e.g., balancedness or sign preservation properties. See Lemma D.1 for details.

---

[1] We note, however, that all arguments in this paper holds for any exponentially-tailed loss function (Soudry et al., 2018).

Note that controlling FLS via the initialization scale factor $\alpha$ is essentially equivalent to using an output multiplier, as in Section 3. We formally show this point in Proposition B.1.

**Data model.** As the data-generating distribution, we consider a simple Gaussian mixture in $\mathbb{R}^d$ with two classes. Precisely, each sample $(\mathbf{x}_i, y_i)$ is generated as

$$\mathbf{x}_i = \kappa y_i \mathbf{s}_i + \sigma \mathbf{z}_i, \qquad \mathbf{z}_i \sim \mathcal{N}(\mathbf{0}, \mathbf{I}_d), \tag{6}$$

where $\mathbf{s}_i \in \{\mathbf{s}_+, \mathbf{s}_-\}$ is the signal vector with $\|\mathbf{s}_i\| = 1$, chosen according to the corresponding class label. The separability parameter $\kappa \in \mathbb{R}_+$ and the noise level $\sigma \in \mathbb{R}_+$ control the signal strength and the noise magnitude, respectively. Note that in our theoretical analyses, we set $\kappa = 1$ and consider symmetric Gaussian mixture for simplicity, i.e., $\mathbf{s}_+ = \mathbf{s}_-$.

We further assume that our training dataset satisfies the following condition (Phuong & Lampert, 2021).

**Assumption 4.1** (Orthogonal separability)**.** There exists some constant $\lambda \in \mathbb{R}_+$ such that for all distinct pairs of training data $(\mathbf{x}, y), (\tilde{\mathbf{x}}, \tilde{y}) \in D$, the following holds:

$$\frac{y\tilde{y}\langle \mathbf{x}, \tilde{\mathbf{x}} \rangle}{\|\mathbf{x}\| \|\tilde{\mathbf{x}}\|} \geq \lambda. \tag{7}$$

Assumption 4.1 is a sufficient condition under which the early-phase ODE admits an *interpretable* stationary point (see Lemma 5.1), a property that does not hold for general datasets (Glasgow, 2024; Boursier & Flammarion, 2025). Since our primary focus is the generalization, we impose this assumption only on the training set, and not the population distribution. Here, we can ensure that Assumption 4.1 holds with high probability in our data model (Proposition E.1).

## 4.3. Key definitions

Phuong & Lampert (2021) and Min et al. (2024) show that there exists some *trapping time*

$$t_1 = O(\log n / \sqrt{\lambda}) \tag{8}$$

independent of the scale factor $\alpha$, after which every neuron becomes permanently specialized to a single class or becomes dead. Formally, consider the *data-dependent cones*

$$\mathcal{C}_+ := \{\mathbf{w} : \mathbb{1}[\langle \mathbf{w}, \mathbf{x}_i \rangle > 0] = \mathbb{1}[y_i > 0], \forall i\}, \tag{9}$$

$$\mathcal{C}_\oslash := \{\mathbf{w} : \langle \mathbf{w}, \mathbf{x}_i \rangle \leq 0, \forall i\}, \tag{10}$$

corresponding to neurons that activate only on the positive class, or never activated, respectively. We can also define $\mathcal{C}_-$ analogously. At time $t_1$, we can partition the indices of the neurons of the given model by which cone they belong to. More formally, we define *neuron index partition* as

$$V_+ := \{j \in [h] : \mathbf{w}_j(t_1) \in \mathcal{C}_+\}, \tag{11}$$

$$V_\oslash := \{j \in [h] : \mathbf{w}_j(t_1) \in \mathcal{C}_\oslash\}, \tag{12}$$

where $V_-$ can be defined analogously.

Min et al. (2024) shows that this partition remains the same for all $t \geq t_1$. Thus, for any such $t$, one can analyze class-wise learning dynamics by decoupling neurons into linear subnetworks indexed by the positive class $V_+$ and the negative class $V_-$ ($V_\oslash$ does not affect training).

In what follows, we focus only on the *positive-class data*, for $t \geq t_1$. The negative class can be handled similarly,[2] and we are not interested in $t < t_1$ as we are interested in the generalization of models that achieve low training risk.

Now, we can define the effective predictor as follows.

**Definition 4.2** (Effective predictor). For $t \geq t_1$, the effective (linear) predictor for the positive-class is defined as

$$\hat{\mathbf{w}}_\alpha(t) := \sum_{j \in V_+} v_j(t)\mathbf{w}_j(t) \qquad (13)$$

Here, the activation function is linear as each neuron is activated only for the data from its corresponding class.

Given this effective predictor, we are interested in the angular alignment (i.e., normalized inner product) between the direction of the effective predictor and the class mean. More concretely, consider the following definition.

**Definition 4.3** (Angular alignment). For $t \geq t_1$, the angular alignment between the effective predictor and some reference direction $\mathbf{r} \in \mathbb{R}^d$ is defined as

$$\Psi(t) := \left\langle \frac{\hat{\mathbf{w}}_\alpha(t)}{\|\hat{\mathbf{w}}_\alpha(t)\|}, \mathbf{r} \right\rangle \qquad (14)$$

In particular, we are interested in analyzing the angular alignment where the reference direction is the class mean

$$\mathbf{r} = \mathbf{x}_+/\|\mathbf{x}_+\|, \quad \text{where} \quad \mathbf{x}_+ := \sum_{i:y_i=+1} \mathbf{x}_i. \qquad (15)$$

In Section 5, we derive a lower bound on this angular alignment, as a function of the scale factor $\alpha$.

## 5. Theoretical Analysis

Based on the formulation described in Section 4, we now provide our main theoretical results. We first provide lower bounds in the neuron alignment during two distinct phases of training (Lemmas 5.2 and 5.4). Then, based on the results we provide an upper bound on the excess error (Theorem 5.6). All proofs in this section are deferred to the Appendix.

### 5.1. Neuron alignment in phase 1

Under the setup specified in Section 4, Min et al. (2024) provides the weight space ODE that governing the Phase 1.

**Lemma 5.1** (Lemma 3 and 4 of Min et al. (2024), informal). *Suppose that the scale factor satisfies*

$$\alpha \leq 1/4\sqrt{h}\mathbf{x}_{\max}\mathsf{W}_{\max}^2. \qquad (16)$$

*Then, for any $t \leq t_\alpha$ (where $t_\alpha \geq t_1$), the alignment ODE (Equation (144)) holds. The stationary point of Equation (144) is given by $\mathbf{x}_+/\|\mathbf{x}_+\|$.*
*(A more formal statement can be found in Lemma D.3.)*

Here, we are particularly interested in the behavior of the alignment at a scale-dependent critical timestamp $t_\alpha = \Theta(\log(1/\alpha)/n)$, which marks the time threshold up to which Lemma 5.1 holds. To analyze this, we first derive the angular alignment between the normalized first-layer weight vector $\mathbf{w}_j(t)/\|\mathbf{w}_j(t)\|$, where $j \in V_+$, and the normalized class mean $\mathbf{x}_+/\|\mathbf{x}_+\|$:

$$\psi_j(t_\alpha) := \left\langle \frac{\mathbf{w}_j(t_\alpha)}{\|\mathbf{w}_j(t_\alpha)\|}, \frac{\mathbf{x}_+}{\|\mathbf{x}_+\|} \right\rangle. \qquad (17)$$

In turn, following results provide a lower bound on $\psi_j(t_\alpha)$ and reveals how it depends on $\alpha$.

**Lemma 5.2.** *For any $j \in V_+$, we have*

$$\psi_j(t_\alpha) \geq \sqrt{\zeta(\alpha)} \tanh\left((t_\alpha - t_1)\|\mathbf{x}_+\|\sqrt{\zeta(\alpha)}\right), \quad (18)$$

*and consequently,*

$$\Psi(t_\alpha) \geq \sqrt{\zeta(\alpha)} \tanh\left((t_\alpha - t_1)\|\mathbf{x}_+\|\sqrt{\zeta(\alpha)}\right), \quad (19)$$

*where $\zeta(\alpha) := 1 - 4\alpha n\sqrt{h}\mathbf{x}_{\max}^2\mathsf{W}_{\max}^2/\|\mathbf{x}_+\|$.*

**Corollary 5.3.** *Suppose that $\|\mathbf{x}_+\|/n < 4\mathbf{x}_{\max}$ holds. Then, the angle between $\mathbf{x}_+$ and $\mathbf{w}_j(t_\alpha)$ is proportional to $\sqrt{\alpha}$.*

As shown in Lemma 5.2 and Corollary 5.3, in the vanishing scale limit (i.e., $\alpha \to 0$), the alignment becomes stronger, for both weights and the effective predictor, with the angular deviation approaching zero. Conversely, as the scale increases, the direction of $\mathbf{w}_j$ deviates more significantly from the class mean. However, this phase does not capture the behavior at reasonable convergence, since the loss has not yet decreased significantly at this phase (Min et al., 2024). Building on these results, we proceed to Phase 2.

### 5.2. Evolution of the alignment in phase 2

Now, we analyze how the results from the phase 1 affect the subsequent training. In particular, we consider the non-asymptotic case where we continue training until the (positive-class) training risk reaches some designated threshold $\eta > 0$. Precisely, we define the *stopping time*[3] as

$$t_{\eta,\alpha} := \inf\{t \geq t_\alpha : \hat{L}_+(\theta_t) \leq \eta\}, \qquad (20)$$

---

[2]See, for example, Min et al. (2024, Section 3)

[3]Although we use the training risk as the stopping criterion, our empirical results still hold when the validation risk is used as the stopping criterion; see Section A.3.

where $\hat{L}_+(\cdot)$ denotes the training risk computed only on positive-class samples. Due to the decoupling of the neurons, the training risk can be decomposed as $\hat{L}(\theta_t) = \hat{L}_+(\theta_t) + \hat{L}_-(\theta_t)$, where each class-wise risk affects only the corresponding subnetwork. Thus, for simplicity, we will write $\hat{L}(\theta) = \hat{L}_+(\theta)$ in what follows. In the same spirit, we will replace $n$ with $n_+ = \sum_i \mathbb{1}[y_i = +1]$, since this modification does not affect the results.

The reason why we consider such $t_{\eta,\alpha}$ is twofold: (1) This choice closely aligns with the common practice, e.g., early stopping; (2) It allows us to go beyond the well-known optimization-generalization trade-off to investigate whether stronger feature learning hurts generalization (Woodworth et al., 2020). Specifically, we examine the behavior at comparable training risk $\eta$ (i.e., at different GF timesteps) for various initialization scales, challenging the view that small initialization is universally beneficial for generalization.

Note that the Phase 2 dynamics are driven by (or, more precisely, initiated by) the result of Lemma 5.2; consequently, the behavior at $t_\eta$ depends on Lemma 5.2. Utilizing such results, we derive the lower bound on $\psi_j(t_{\eta,\alpha})$.

**Lemma 5.4.** *Let $\beta := \lambda^2 \mathbf{x}_{\min}^2 / 32\mathbf{x}_{\max}$, where $\mathbf{x}_{\min}$ denotes the minimum value of all $\|\mathbf{x}\|$. Also let $t_2 \geq t_\alpha = O(\log(1/\alpha)/n)$. Then, for any threshold $\eta > 0$, we have*

$$\psi_j(t_{\eta,\alpha}) \geq \lambda + m(\alpha)\exp(-g(\alpha)), \qquad (21)$$

*where $m(\alpha) := \psi_j(t_\alpha) - \lambda$ and*

$$g(\alpha) \leq \mathbf{x}_{\max} n \left( (t_2 - t_\alpha)\hat{L}(t_\alpha) + \frac{1}{\beta}\log\frac{\hat{L}(t_2)}{\eta} \right).$$

Here, same as in Equation (19) of Lemma 5.2, we can derive the same lower bound on $\Psi(t_{\eta,\alpha})$ using properties of the conic hull, a derivation we omit for brevity.

From Lemma 5.2, the results imply that $m(\alpha)$ in Equation (21) increases as $\alpha$ decreases. In contrast, $g(\alpha)$ depends on a non-asymptotic timescale, which makes it difficult to interpret directly. Nevertheless, we can analyze it indirectly: Since $t_2$ is defined as the timescale at which the loss decreases *significantly* (Min et al., 2024), we expect $t_2 - t_\alpha \approx 0$. Consequently, we may (approximately) bound $g(\alpha) \lesssim \mathbf{x}_{\max} n(\log(\hat{L}(t_2)/\eta)/\beta) \approx O(1)$. Plugging this estimate into Equation (21) suggests that $\exp(-g(\alpha)) \approx 1$, which yields

$$\Psi(t_{\eta,\alpha}) \approx \Psi(t_\alpha). \qquad (22)$$

We validate this empirically and present results in Section E.2, with further discussions. There, we observe that the results aligns with our analysis. Equation (22) suggests that the alignment is *almost consistent* in the phase 2, as they mainly follow the results from the phase 1.

**Comparison with prior work.** Prior works on implicit bias in (deep) linear classification show that the predictor converges asymptotically to the $\ell_2$ max-margin direction of the training set, which is considered desirable for linearly separable datasets (e.g., the hard-margin SVM solution) (Gunasekar et al., 2018; Ji & Telgarsky, 2019; Yun et al., 2021; Phuong & Lampert, 2021; Min et al., 2025). In contrast, we analyze how much the effective predictor can deviate from the reference direction after a finite number of GF iterations; this perspective is particularly useful for analyzing generalization in Gaussian mixtures.

### 5.3. Over-alignment vs. Over-fitting

So far, we have analyzed the alignment of the neurons (and the effective predictor) to the class mean direction, dependent on the scale factor $\alpha$. In this subsection, we connect these results to provide an upper bound on the population error. Our main result (Theorem 5.6) reveals the pitfalls of overly small initialization, which we term *over-alignment*.

We begin by defining the population error as follows.

**Definition 5.5** (Population error). The (zero-one) population error rate of a predictor $\hat{\mathbf{w}}_\alpha \in \mathbb{R}^d$ is

$$\mathcal{E}(\hat{\mathbf{w}}_\alpha) := \Pr\left(\mathrm{sgn}\left(\hat{\mathbf{w}}_\alpha^\top \mathbf{x}\right) \neq y\right), \qquad (23)$$

where the $\Pr(\cdot)$ denotes the probability with respect to the data distribution described in Equation (6).

Let us denote the Bayes optimal error achievable for the same dataset as $\mathcal{E}^*$. Then, we can decompose the excess error of the given predictor $\hat{\mathbf{w}}_\alpha$ into two terms.

$$\mathcal{E}(\hat{\mathbf{w}}_\alpha) - \mathcal{E}^*$$
$$= \underbrace{\inf_{\mathbf{v} \in H(\alpha)} \mathcal{E}(\mathbf{v}) - \mathcal{E}^*}_{=:\mathsf{OA}(\alpha)} + \underbrace{\mathcal{E}(\hat{\mathbf{w}}_\alpha) - \inf_{\mathbf{v} \in H(\alpha)} \mathcal{E}(\mathbf{v})}_{=:\mathsf{OF}(\alpha)}. \qquad (24)$$

Here, the set $H(\alpha)$ denotes the circular cone around the one-side class mean, characterizing the region where the effective predictor resides after the time $t_{\eta,\alpha}$ has elapsed:

$$H(\alpha) = \{\mathbf{v} \in \mathbb{S}^{d-1} : \langle \mathbf{x}_+/\|\mathbf{x}_+\|, \mathbf{v}\rangle \geq \Psi(t_{\eta,\alpha})\}. \quad (25)$$

Note that we have constrained the $\ell_2$ norm of $\mathbf{v}$ to be one. This is because the zero-one error is invariant to scalar multiplication of the weight, which simplifies the analysis.

In Equation (24), we have introduced two terms, $\mathsf{OA}(\alpha)$ and $\mathsf{OF}(\alpha)$, which we refer to as the degrees of *over-alignment* and *over-fitting*, respectively. Intuitively, each term can be interpreted as follows:

- $\mathsf{OA}(\alpha)$, which we dubbed *over-alignment*, is the gap between the minimum achievable error among all predictors in $H(\alpha)$ and the Bayes error. As we will show below, this quantity **decreases as $\alpha$ increases**.

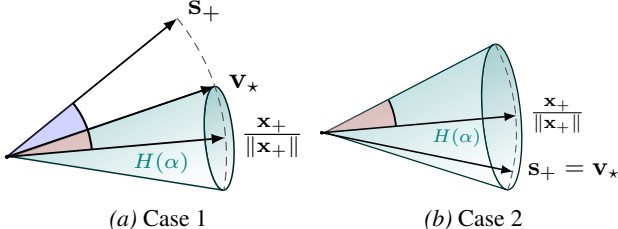

*(a)* Case 1  *(b)* Case 2

*Figure 4.* **Visual explanation.** In our analysis, $\mathsf{OA}(\alpha)$ is determined by the angle, whereas $\mathsf{OF}(\alpha)$ is determined by the angle. (a) Case 1: When $\alpha$ is sufficiently small, there exists an irreducible gap between $\mathbf{s}_+$ and $\mathbf{v}_\star \in H(\alpha)$. In this case, the error is determined by both $\mathsf{OA}(\alpha)$ and $\mathsf{OF}(\alpha)$. (b) Case 2: When $\alpha$ is sufficiently large, we have $\mathbf{s}_+ = \mathbf{v}_\star$. In this case, the error is governed by the "volume" of $H(\alpha)$, i.e., by $\mathsf{OF}(\alpha)$ alone.

- $\mathsf{OF}(\alpha)$, referred to as *over-fitting*, is the gap between the population error of the learned predictor $\hat{\mathbf{w}}_\alpha$ and the minimum achievable error within $H(\alpha)$. We will show below that this quantity **increases as $\alpha$ increases**.

The theorem below provides an upper bound on the excess error of the learned predictor, where the bound characterizes the trade-off between $\mathsf{OA}(\alpha)$ and $\mathsf{OF}(\alpha)$. Below, for simplicity, we write $\phi := \angle(\mathbf{x}_+, \mathbf{s}_+)$.

**Theorem 5.6.** *Suppose that Assumption 4.1 holds, $\|\hat{\mathbf{w}}_\alpha(t_{\eta,\alpha})\| \leq 1$, and $\langle \hat{\mathbf{w}}_\alpha(t_{\eta,\alpha}), \mathbf{x}_i \rangle \geq 0$ for all $i$ with $y_i = +1$. Let $G_\epsilon$ be the grid defined by $G_\epsilon := \{-1 + k\epsilon : k \in \mathbb{Z}\} \cap [-1, 1]$. Then, for any $\delta \in (0, 1)$ and $\epsilon \in (0, 0.25)$, we have*

$$\mathcal{E}(\hat{\mathbf{w}}_\alpha(t_{\eta,\alpha})) - \mathcal{E}^* \leq \underbrace{\Phi\left(-\frac{\mathbf{v}_\star^\top \mathbf{s}_+}{\sigma}\right) - \Phi\left(-\frac{1}{\sigma}\right)}_{=\mathsf{OA}(\alpha)} \quad (26)$$

$$+ \underbrace{\frac{2(1+e)}{\sigma\sqrt{2\pi}}\left(g(\alpha)h(n,d) + \eta + C\left(1 + \sigma\sqrt{d}\right)\sqrt{\frac{\log(6/\delta\epsilon)}{n}}\right)}_{\geq \mathsf{OF}(\alpha)}$$

*with probability at least $1 - \delta$, where:*

$$\mathbf{v}_\star := \arg\max_{\mathbf{v} \in H(\alpha)} \mathbf{v}^\top \mathbf{s}_+, \quad (27)$$

$$g(\alpha) := 2\sqrt{2\pi} \cdot \sqrt{1 - r(\alpha)^2}, \quad (28)$$

$$h(n,d) := \sqrt{\frac{d}{n}}\left(\sigma\sqrt{\frac{d}{n}} + \sigma + 1\right), \quad (29)$$

$$r(\alpha) := \max\left\{r \in G_\epsilon : 0 < r \leq \min_{\mathbf{v} \in H(\alpha)} \mathbf{v}^\top \mathbf{s}_+\right\} \quad (30)$$

*and for some constants $C > 0$.*

To understand what Theorem 5.6, let us take a closer look at the dependencies of $\mathsf{OA}(\alpha)$ and $\mathsf{OF}(\alpha)$ on $\alpha$.

**Over-alignment.** For $\mathsf{OA}(\alpha)$, there exists two regimes. First, suppose that the scale factor $\alpha$ is sufficiently large (while

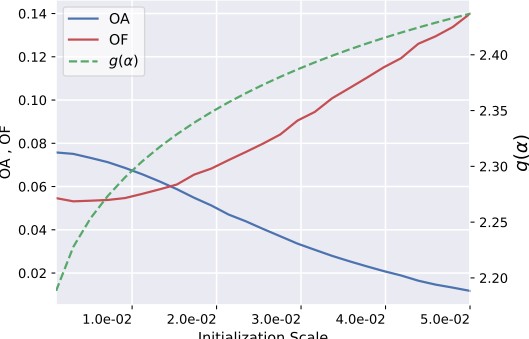

*Figure 5.* **Numerical simulation of $\mathsf{OA}(\alpha)$, $\mathsf{OF}(\alpha)$, and $g(\alpha)$:** Note that $\mathsf{OA}(\alpha) + \mathsf{OF}(\alpha)$ recovers the excess error.

still satisfying the condition from Lemma 5.1), so that $\Psi(t_{\eta,\alpha}) \leq \cos\phi$ holds. Then, we know that $\mathbf{v}_\star = \mathbf{s}_+$ holds and thus the over-alignment term becomes equal to zero (e.g., Case 2 in Figure 4).

In the second regime, we consider the case where $\alpha$ is smaller than this threshold. Then, for moderate number of data $n$ and data dimension $d$, we know that $\mathsf{OA}(\alpha)$ is a non-increasing function of $\alpha$. Furthermore, let $\bar{\Psi}(\cdot) = \arccos\Psi(\cdot)$, then we have

$$\mathbf{v}_\star = \frac{\sin(\phi - \bar{\Psi}(t_{\eta,\alpha}))}{\|\mathbf{x}_+\|\sin\phi}\mathbf{x}_+ + \frac{\sin(\bar{\Psi}(t_{\eta,\alpha}))}{\sin\phi}\mathbf{s}_+, \quad (31)$$

which corresponds to the spherical linear interpolation (e.g., Case 1 in Figure 4).

Together, these results demonstrate the phenomenon which we call *over-alignment*: When $\alpha$ is small, there is a generalization gap due to the discrepancy between the best achievable predictor in the cone $H(\alpha)$ (i.e., $\mathbf{v}_\star$) and the Bayes optimal predictor (i.e., $\mathbf{s}_+$). This gap becomes more severe as $\alpha$ decreases, leading to an increase in generalization error in the large FLS regime.

**Over-fitting.** We now turn our attention to $\mathsf{OF}(\alpha)$. Since the exact equality for $\mathsf{OF}(\alpha)$ cannot be derived (due to $\mathcal{E}(\hat{\mathbf{w}}_\alpha)$ term), we upper bound the term as in Equation (26): Here, we study the behavior of $g(\alpha)$, since the only $\alpha$-dependent term among three terms in $\mathsf{OF}(\alpha)$.

Here, the term $r(\alpha) \approx \min_{\mathbf{v} \in H(\alpha)} \mathbf{v}^\top \mathbf{s}_+$ captures the geometry of the cone $H(\alpha)$ (more precisely, the angle of the cone). For instance, when $n$ and $d$ are fixed and $\alpha$ is sufficiently small (e.g., Case 1 in Figure 4), $r(\alpha)$ increases, hence $g(\alpha)$ decreases. Otherwise, for large $\alpha$, we have an increased $g(\alpha)$ (e.g., Case 2 in Figure 4).

We refer to this term—represented by $g(\alpha)$—as *over-fitting*, since this phenomenon is consistent with the traditional notion of over-fitting in learning theory: As $\alpha$ increases, the volume of the hypothesis space of the learned predictor (i.e., $H(\alpha)$) also increases, leading to poor generalization.

**Numerical experiments.** To validate our theoretical results, in Figure 5, we plot $\mathsf{OA}(\alpha)$, $\mathsf{OF}(\alpha)$, and $g(\alpha)$. We observe that $\mathsf{OA}(\alpha)$ and $\mathsf{OF}(\alpha)$ exhibit a trade-off, yielding an optimal FLS for the excess error (see Section A.7). Also, the trend of estimated $g(\alpha)$ correlates well with $\mathsf{OF}(\alpha)$. These results suggest that, even in large FLS regime, we can clearly formalize the optimal FLS. For details, see Section A.6.

**Small-norm regime.** We also note that Theorem 5.6 holds in the small-norm regime, i.e., $\|\hat{\mathbf{w}}(t_{\eta,\alpha})\| \leq 1$, in contrast to most prior work on implicit bias in classification, which studies the norm-exploding regime where $\|\hat{\mathbf{w}}(t_{\eta,\alpha})\| \rightarrow \infty$; see, e.g., Soudry et al. (2018). Since we are mainly interested in (1) small $\alpha$ and (2) finite-time training, this condition can be deemed realistic (e.g., see Figure 17).

**Asymptotic analysis.** One might ask how these phenomena appears as $d$ and $n$ change. Unlike our work—which focuses on the finite-sample/dimensional regime—several prior works on Gaussian mixture classification have investigated the generalization error particularly in the *proportional asymptotic regime*, where $d, n \rightarrow \infty$ with $d/n \in (0, \infty)$ (Mignacco et al., 2020; Refinetti et al., 2021). Applying the similar idea to our setting, under the modified proportional limit (induced by Assumption 4.1), we can obtain the following results.

**Proposition 5.7.** *Let* $\gamma_1 := d/\left(n^2 \log n\right)$ *and* $\gamma_2 := \kappa^2/\left(\sigma^2 \sqrt{d \log n}\right)$. *Suppose* $d, n \rightarrow \infty$ *and* $\gamma_2 \rightarrow \gamma_{2,\infty} \in (0, \infty)$. *Then, with probability tending to one, we have*

$$\tan^2 \phi \rightarrow \sqrt{\gamma_1}/\gamma_2. \tag{32}$$

*Consequently, we can divide the regimes as follows:*

1. *Data-abundant: If $\gamma_1 \rightarrow 0$, then $\phi \rightarrow 0$.*

2. *Moderate: If $\gamma_1 \rightarrow \gamma_{1,\infty} \in (0, \infty)$, then $\phi \rightarrow \arctan\left(\gamma_{1,\infty}^{1/4}/\gamma_{2,\infty}^{1/2}\right)$.*

3. *High-dimensional: If $\gamma_1 \rightarrow \infty$, then $\phi \rightarrow \pi/2$.*

From Proposition 5.7, we can notice that the trade-off between over-alignment and over-fitting arises only in the "moderate" regime, since $\phi$ converges to a nonzero angle between 0 and $\pi/2$: For example, in the "data-abundant" regime, the empirical mean recovers the population signal direction—i.e., $\mathbf{x}_+/\|\mathbf{x}_+\| = \mathbf{s}_+$—and the optimal $\alpha$ converges to zero. Hence, the emergence of an optimal FLS can be viewed as a byproduct of the practical training regime, namely the finite-sample and finite-dimensional setting.

### 5.4. Transferring the Optimal FLS

Can we leverage this phenomenon in practice? Indeed, recent works in deep learning theory have shown that, in some cases, optimal hyperparameters (HPs) can be transferred across different architectural configurations, making HP tuning more efficient at scale (Yang et al., 2021; Mlodozeniec

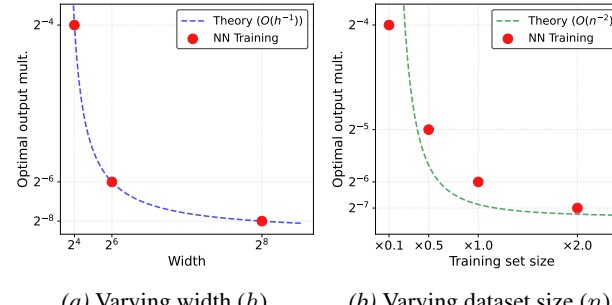

*(a)* Varying width $(h)$       *(b)* Varying dataset size $(n)$

*Figure 6.* **Optimal FLS (output multiplier) is predictable across width & dataset size.** The dotted line indicates the scaling predicted by our theory, while the red dots represent the values obtained from training.

et al., 2026). Motivated by this perspective, in 5-layer vanilla CNNs, we show that the optimal FLS—as an instance of HPs—is transferable across widths and training dataset size; see Section A.2 for experimental details.

In Figures 6a and 6b, we plot the optimal output multiplier $c_\star$ for generalization by varying the width and training set size, respectively. Here, the results suggest that $c_\star$ are closely aligns with a numerical scaling law with respect to each factor. We note that these results align closely with our theory, which predicts that the $c_\star \propto O(n^{-2}h^{-1})$, obtained from differentiating the error bound. Extending the transfer argument beyond simple networks requires understanding the nontrivial effects arising from various factors, thus we leave this as future work.

## 6. Conclusion

In this work, we study how feature learning strength (FLS) affects generalization in classification tasks. Empirically, we find that not only small FLS, but also extremely large FLS can hurt generalization in deep networks, leading to a U-shaped curve in generalization performance (e.g., test accuracy or loss). To understand this phenomenon theoretically, we investigate its origin through the notions of *over-alignment* and *over-fitting*, which are obtained by decomposing the excess error based on the finite-time training dynamics of two-layer ReLU networks. We show that these two quantities exhibit a trade-off as a function of FLS, and that the optimal FLS arises from balancing them.

**Limitation and future direction.** Our results do not capture the effects of many techniques used in practice, such as stochastic and adaptive gradient methods, data augmentation, and so on. Additionally, our theoretical framework relies on a strict constraint on the training dataset (i.e., orthogonal separability); Relaxing the assumption and extending the analysis to more practical regimes would be a important future work. From a practical perspective, one promising direction is to develop a rigorous framework for analyzing the effect of FLS in larger models (e.g., Transformers).

## Acknowledgements

This work was supported by Institute of Information & Communications Technology Planning & Evaluation (IITP) grant funded by the Korea government (MSIT) (No.RS-2024-00457882, No.RS-2019-II191906, No.RS-2022-II220713), the National Research Foundation of Korea (NRF) grant funded by the Korea government (MSIT) (No.RS-2024-00453301, No.RS-2025-24873016, No.RS-2026-25494004), and Basic Science Research Program through the National Research Foundation of Korea (NRF) funded by the Ministry of Education (No.RS-2025-25421671).

## Impact Statement

This paper presents work whose goal is to advance the field of Machine Learning. There are many potential societal consequences of our work, none which we feel must be specifically highlighted here.

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

# Appendix

# A. Experimental Details and Additional Results

In this section, we provide further experimental details and omitted results. For all training runs, we use a single GPU of NVIDIA RTX 3090/4090 or A6000.

## A.1. Details About Experiments in Section 3

**Training details.** Across all training runs, we use a batch size of 128 and the vanilla SGD optimizer (without momentum). For training iterations, we trained until 80 epochs for all runs, which we found to be enough (i.e., all networks reach the best test accuracy and the lowest test loss within these epochs). To achieve near-perfect training accuracy (and near-zero training loss), we do not use data augmentation. Moreover, we do not use other training techniques, such as a learning rate scheduler or weight decay.

**Dataset details.** Here, we describe specific details for the image datasets used in the experiments.

- **CIFAR-10** and **CIFAR-100** each consist of 50k training images and 10k test images, with 10 and 100 classes, respectively.
- For the **BigGAN-generated dataset**, we generate 1k images per class (i.e., 10k images in total for 10 classes).[4] We then randomly select 8k images for training, ensuring class balance, and use the remaining 2k images as the test set. We choose 10 classes from the dog category of ImageNet: 'basenji,' 'basset,' 'beagle,' 'borzoi,' 'keeshond,' 'standard poodle,' 'vizsla,' 'weimaraner,' 'whippet,' and 'yorkshire terrier.' All images are resized to $32 \times 32$ resolution. We provide example images in Figure 7.

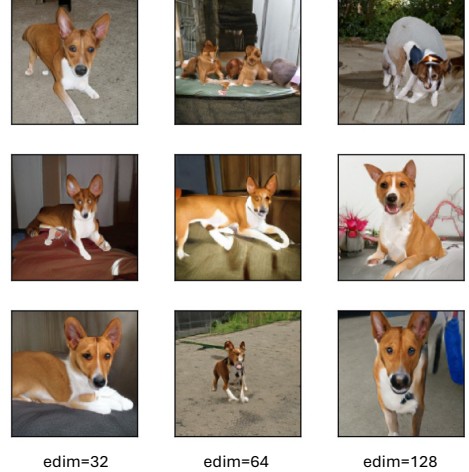

edim=32      edim=64      edim=128

*Figure 7.* **Example images from the BigGAN-generated datasets.** As the effective dimensionality (i.e., edim) of the input increases, BigGAN produces more diverse images, thereby making the task more difficult.

---

[4] We additionally generate 10k images for the experiments in Section 5.4.

### A.2. Details About Experiments in Section 5.4

We use a 5-layer vanilla CNN as the neural network and a BigGAN-generated dataset with an effective dimension of 128 as the training dataset. As in Section 3, we use a batch size of 128 and the vanilla SGD optimizer without momentum. In the width-scaling experiments, we vary the number of channels in the CNN. The optimal output multiplier is defined as the value of the multiplier that minimizes the test loss.

### A.3. Validation risk as stopping criterion

In this subsection, we provide additional experimental results obtained when the stopping criterion is changed to *validation risk*. To this end, we construct a fixed validation set using 20% of the original training set.

The results are shown in Table 1. We observe that the optimal output multiplier does not change, even when the network is early-stopped once the optimal validation loss is achieved.

*Table 1.* **Test accuracy (%) over 80 epochs, with early stopping based on the minimum validation loss.** We train ResNet-18 on a BigGAN-generated dataset (edim=128).

| Output mult. | $2^{-10}$ | $2^{-8}$ | $2^{-6}$ | $2^{-4}$ | $2^{-2}$ | $2^0$ | $2^2$ |
|---|---|---|---|---|---|---|---|
| Peak acc. (80 epochs) | 72.92 | 75.35 | **76.62** | 75.18 | 69.22 | 59.95 | 49.78 |
| ES w/ val. loss | 67.63 | 70.63 | **73.35** | 71.53 | 66.68 | 56.83 | 47.20 |

## A.4. Experiments on CIFARs

In this subsection, we present additional results on CIFAR-10 and CIFAR-100 with varying effective dimensionality, as a follow-up to Figure 2 in the main paper. Specifically, we report the peak test accuracy and the best (i.e., lowest) test loss achieved during training for VGG19 and ResNet{18,34,50}.

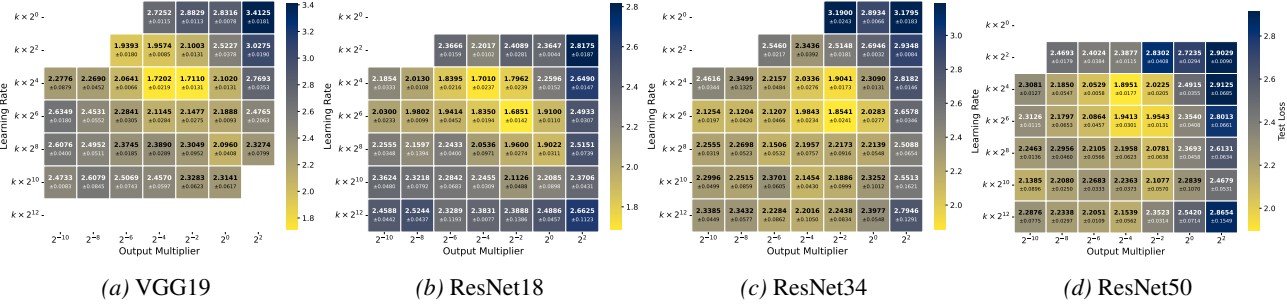

*Figure 8.* Best test loss (CIFAR-100).

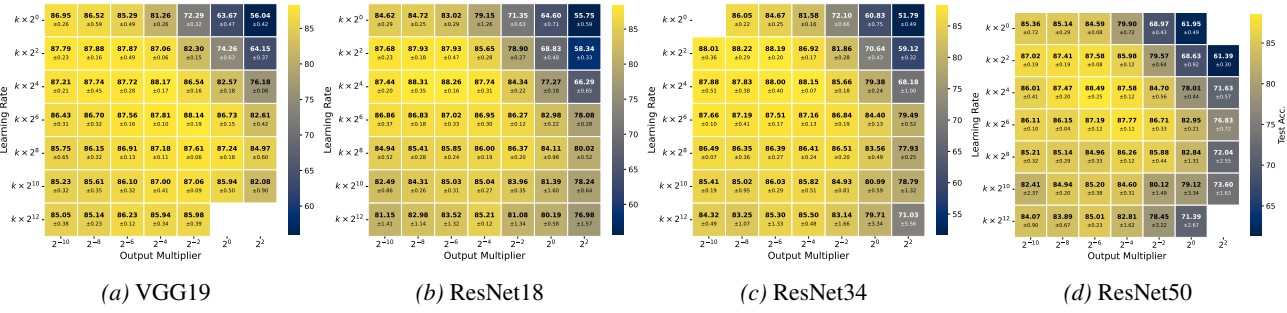

*Figure 9.* Peak test accuracy (CIFAR-10).

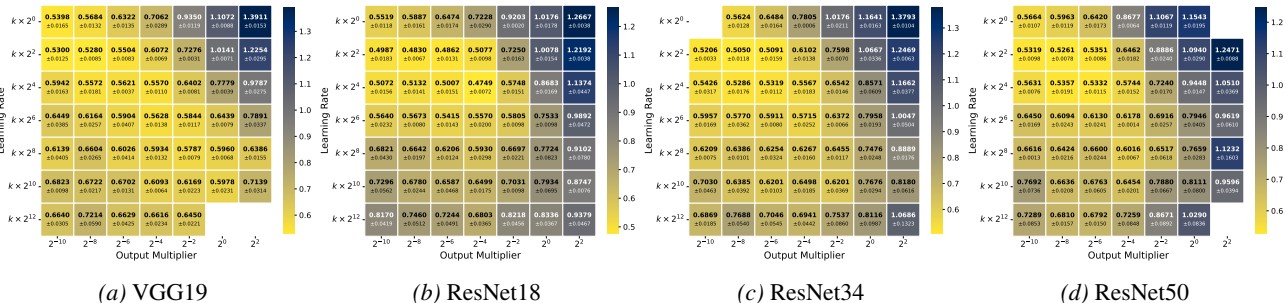

*Figure 10.* Best test loss (CIFAR-10).

## A.5. Experiments on Synthetic Image Datasets

In this subsection, we present additional results on BigGAN-generated dataset with varying effective dimensionality, as a follow-up to Figure 3 in the main paper. Specifically, we report the best test accuracy and the best (i.e., lowest) test loss achieved during training for VGG19 and ResNet18.

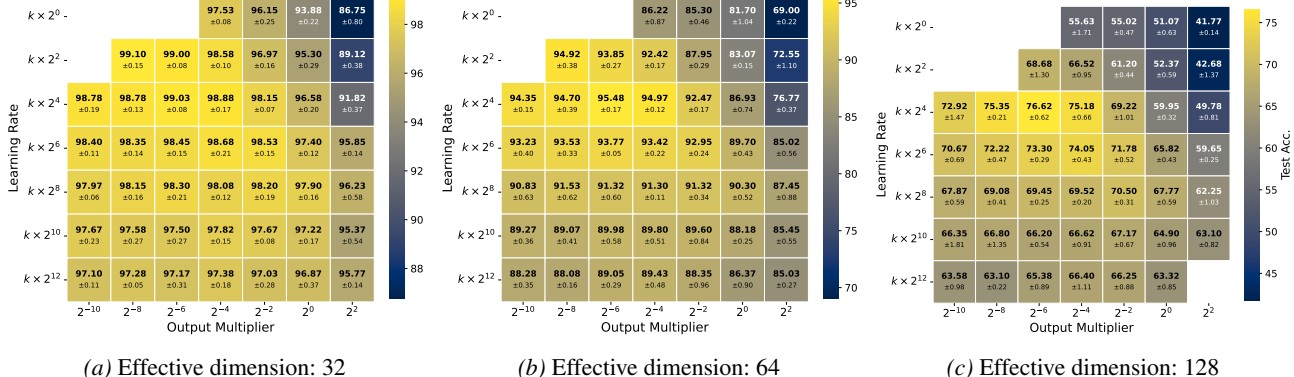

*(a)* Effective dimension: 32      *(b)* Effective dimension: 64      *(c)* Effective dimension: 128

*Figure 11.* **Peak test accuracy for ResNet18** trained on a BigGAN-generated dataset, with varying effective dimensionality.

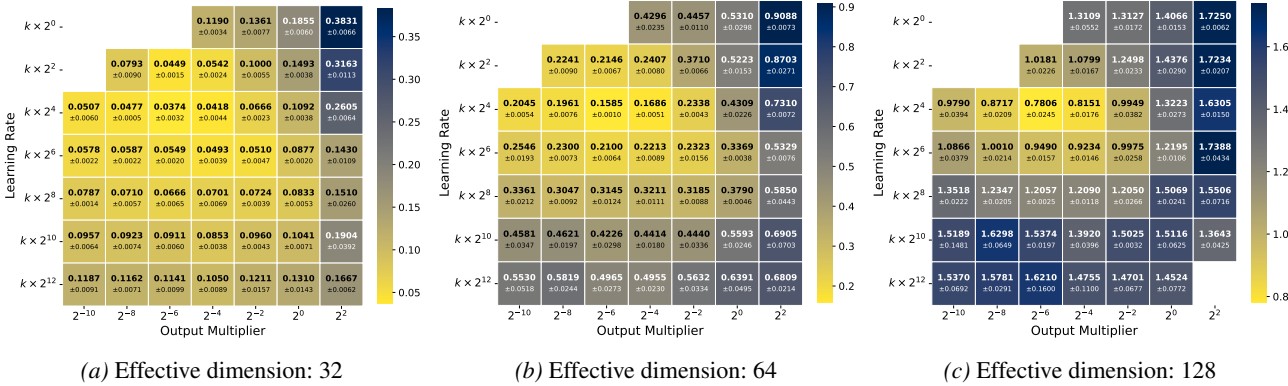

*(a)* Effective dimension: 32      *(b)* Effective dimension: 64      *(c)* Effective dimension: 128

*Figure 12.* **Best test loss for ResNet18** trained on a BigGAN-generated dataset, with varying effective dimensionality.

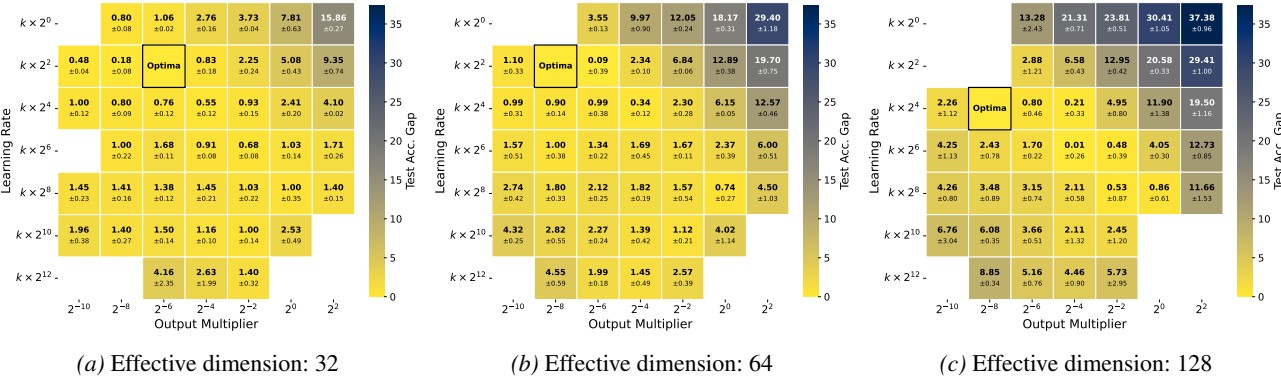

*(a)* Effective dimension: 32      *(b)* Effective dimension: 64      *(c)* Effective dimension: 128

*Figure 13.* **Gap of the peak test for VGG-19** trained on a BigGAN-generated dataset, with varying effective dimensionality.

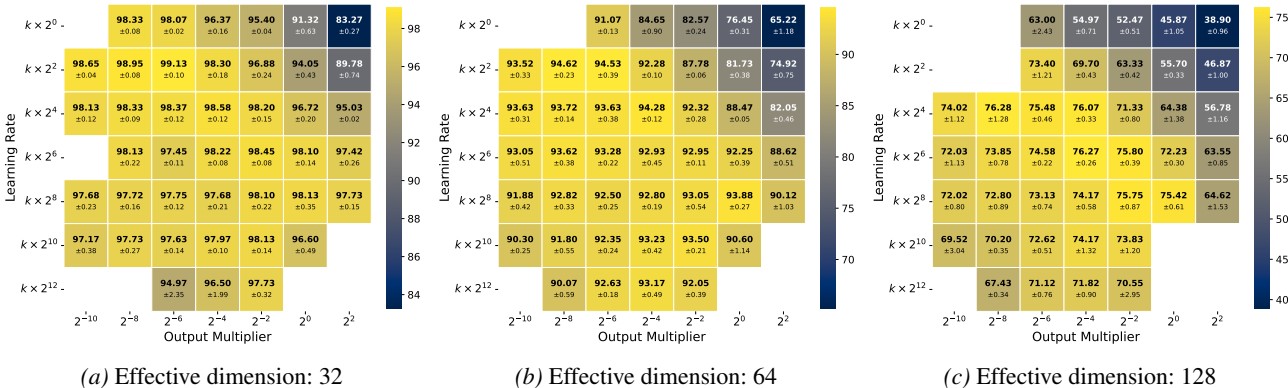

*(a)* Effective dimension: 32      *(b)* Effective dimension: 64      *(c)* Effective dimension: 128

*Figure 14.* **Peak test accuracy for VGG19** trained on a BigGAN-generated dataset, with varying effective dimensionality.

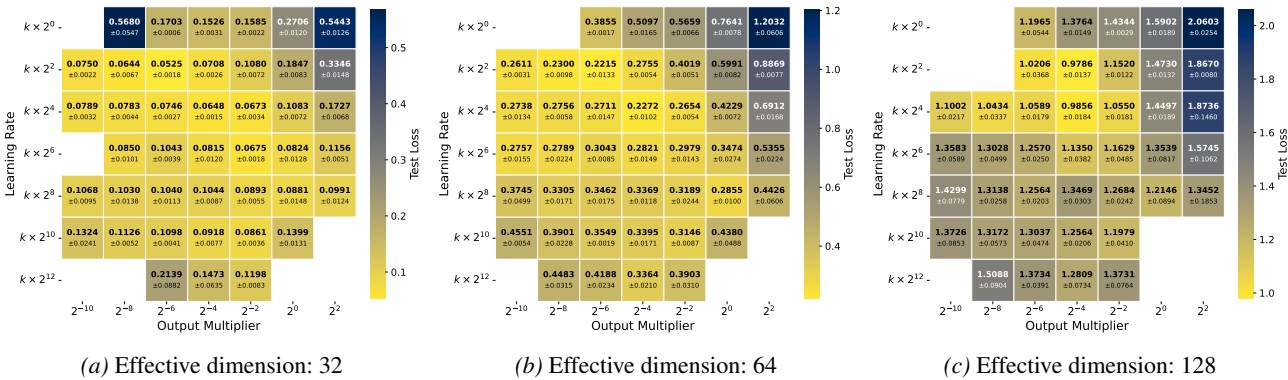

*(a)* Effective dimension: 32      *(b)* Effective dimension: 64      *(c)* Effective dimension: 128

*Figure 15.* **Best test loss for VGG19** trained on a BigGAN-generated dataset, with varying effective dimensionality.

### A.6. Details About Experiments in Section 5

For numerical experiments (Figure 5 and Figure 16), we use 50 training samples generated from a Gaussian mixture model with $\kappa = 1.5$, $\sigma = 1$, and 128 input dimension. We use two-layer, bias-free ReLU networks with 64 hidden units, and train all models until the training error reaches $\eta = 0.05$. Note that we do not restrict $\lambda$ here in order to reflect a realistic setup.

### A.7. Additional Results in Section 5

In this subsection, we plot the excess error (i.e., $\mathsf{OA}(\alpha) + \mathsf{OF}(\alpha)$) from our numerical experiments (Figure 5). As shown in Figure 16, the excess error exhibits a "U-shape," demonstrating the existence of an optimal FLS.

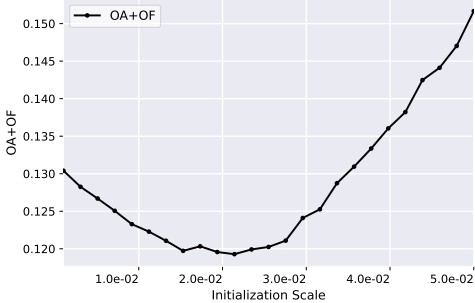

*Figure 16.* Excess error across different initializations scale ($\eta = 0.05$)

**Discussion of the small norm regime.** Our main theorem Theorem 5.6 theoretically requires $\|\hat{\mathbf{w}}_\alpha\| \leq 1$. In Figure 17, we present at finite-time training, norm of the predictor indeed tends to be small. If the initialization scale becomes large, the norm can blow up; however, we do not consider this regime in our theorem. However, empirically, we observe that our analysis still holds even with longer training (which also makes the norm blow up) (Figure 18).

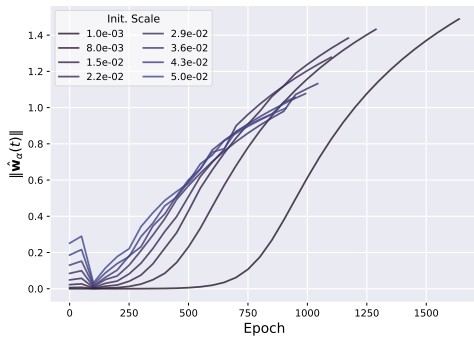

*Figure 17.* Norm of the effective predictor.

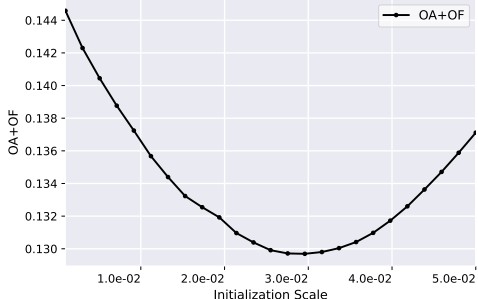

*Figure 18.* Excess error across different initializations scale ($\eta = 0.01$).

# B. Output Scaling vs. Initialization Scaling

In Proposition B.1, we show that output scaling with scale-compensated learning rate (used in Section 3) is exactly equivalent to the initialization scaling (used in Section 4).

**Proposition B.1.** *Suppose* $\mathbf{W} = (\mathbf{W}_1, \dots, \mathbf{W}_L)$ *and* $\mathbf{W}' = (\alpha\mathbf{W}_1, \dots, \alpha\mathbf{W}_L)$ *are the parameters of an L-layer bias-free, positively homogeneous network, where* $\alpha > 0$ *is the initialization scaling factor applied to all layers. Consider two learning configurations* $A_{\mathbf{W}} = (\alpha^L, \eta)$ *and* $A_{\mathbf{W}'} = (1, \eta\alpha^2)$, *where each pair denotes the output multiplier and the learning rate, respectively. Then, under gradient descent (or gradient flow), for all* $t \geq 0$ *and all* $l \in [L]$, *we have* $\mathbf{W}'_l(t) = \alpha\,\mathbf{W}_l(t)$. *In particular,* $f_{A_{\mathbf{W}}}(\mathbf{x}; \mathbf{W}(t)) = f_{A_{\mathbf{W}'}}(\mathbf{x}; \mathbf{W}'(t))$ *for all* $\mathbf{x}$ *and* $t$.

*Proof.* Note that $f_{\mathbf{W}'}(\mathbf{x}) = \alpha^L f_{\mathbf{W}}(\mathbf{x})$. Moreover, for each $l \in [L]$, we have

$$\nabla_{\mathbf{W}'_\ell} f_{\mathbf{W}'}(\mathbf{x}) = (\mathbf{W}'_L \cdots \mathbf{W}'_{\ell+1})(\mathbf{W}'_{\ell-1} \cdots \mathbf{W}'_1 \mathbf{x})^\top = \alpha^{L-1} \nabla_{\mathbf{W}_\ell} f_{\mathbf{W}}(\mathbf{x}). \tag{33}$$

Let $\gamma = \alpha^L$ and define losses $\hat{L}_\gamma(\mathbf{W}) = \frac{1}{n}\sum_{i=1}^{n} \ell(\gamma f_{\mathbf{W}}(\mathbf{x}_i), y_i)$ and $\hat{L}_1(\mathbf{W}') = \frac{1}{n}\sum_{i=1}^{n} \ell(f_{\mathbf{W}'}(\mathbf{x}_i), y_i)$. Since $f_{\mathbf{W}'}(\mathbf{x}_i) = \gamma f_{\mathbf{W}}(\mathbf{x}_i)$, we have $\ell'(f_{\mathbf{W}'}(\mathbf{x}_i), y_i) = \ell'(\gamma f_{\mathbf{W}}(\mathbf{x}_i), y_i)$, and thus

$$\nabla_{\mathbf{W}_\ell} \hat{L}_\gamma(\mathbf{W}) = \frac{1}{n}\sum_{i=1}^{n} \ell'(\gamma f_{\mathbf{W}}(\mathbf{x}_i), y_i) \cdot \gamma \cdot \nabla_{\mathbf{W}_\ell} f_{\mathbf{W}}(\mathbf{x}_i), \tag{34}$$

$$\nabla_{\mathbf{W}'_\ell} \hat{L}_1(\mathbf{W}') = \frac{1}{n}\sum_{i=1}^{n} \ell'(\gamma f_{\mathbf{W}}(\mathbf{x}_i), y_i) \cdot \nabla_{\mathbf{W}'_\ell} f_{\mathbf{W}'}(\mathbf{x}_i) = \frac{1}{\alpha}\nabla_{\mathbf{W}_\ell} \hat{L}_\gamma(\mathbf{W}). \tag{35}$$

With $\eta' = \eta\alpha^2$, the GD update gives

$$\mathbf{W}'_\ell(t+1) = \mathbf{W}'_\ell(t) - \eta'\nabla_{\mathbf{W}'_\ell}\hat{L}_1(\mathbf{W}'(t)) \tag{36}$$

$$= \alpha\mathbf{W}_\ell(t) - \eta\alpha^2 \cdot \frac{1}{\alpha}\nabla_{\mathbf{W}_\ell}\hat{L}_\gamma(\mathbf{W}(t)) \tag{37}$$

$$= \alpha\mathbf{W}_\ell(t+1), \tag{38}$$

and the same calculation applies to gradient flow. Since $\mathbf{W}'(0) = \alpha\mathbf{W}(0)$, induction yields $\mathbf{W}'_\ell(t) = \alpha\mathbf{W}_\ell(t)$ for all $t \geq 0$ and $\ell \in [L]$, hence $f_{A_{\mathbf{W}}}(\mathbf{x}; \mathbf{W}(t)) = f_{A_{\mathbf{W}'}}(\mathbf{x};, \mathbf{W}'(t))$ for all $\mathbf{x}$ and $t$, and this completes the proof. $\square$

**Remark B.2.** For $L = 2$, analyzing $A_{\mathbf{W}} = (\alpha^2, \eta/\alpha^2)$ is equivalent to analyzing $A_{\mathbf{W}'} = (1, \eta)$, which matches our setup. Note that in Proposition B.1, the distribution of each layer's weights is unconstrained. In our manuscript, we consider a setup where the distribution of the second layer weights is conditioned on the sampled first layer weights, which is included as a special case of Proposition B.1.

## B.1. Feature Learning Strength and Maximal Update Parametrization

Maximal update parametrization (Yang & Hu, 2021), also known as $\mu$P, controls the FLS through the output multiplier $a := (b\sqrt{\text{width}})^{-1}$. Since scalar $b > 0$ is a *free variable* (see Karkada (2024) for more details), this can equivalently be rewritten in terms of the output multiplier $c = (b\sqrt{\text{width}})^{-1}$ as in our experiments. Therefore, the two formulations ultimately capture the same underlying notion.

## C. Proofs in Section 4

In this section, we write $\mathcal{I}_+ := \{i \in [n] : y_i = +1\}$.

### C.1. Proof of Lemma 5.2

Consider the quantity $\mathbf{x}_a(\mathbf{w}) := \sum_{i:\langle \mathbf{x}_i, \mathbf{w} \rangle > 0} y_i \mathbf{x}_i$. Then, for all $t \in [t_1, t_\alpha]$, we know that

$$\mathbf{x}_a(\mathbf{w}_j(t)) = \sum_{i:\langle \mathbf{x}_i, \mathbf{w}_j(t) \rangle > 0} y_i \mathbf{x}_i = \sum_{i \in \mathcal{I}_+} \mathbf{x}_i = \mathbf{x}_+. \tag{39}$$

Hence, $\langle \mathbf{x}_+/\|\mathbf{x}_+\|, \mathbf{x}_a(\mathbf{w}_j)/\|\mathbf{x}_a(\mathbf{w}_j)\| \rangle = 1$. Then, by Min et al. (2024, Lemma 10), we have

$$\left| \frac{d}{dt} \psi_j(t) - \left(1 - \psi_j^2(t)\right) \|\mathbf{x}_+\| \right| \leq 2n\mathbf{x}_{\max} \max_i |f(\mathbf{x}_i; \mathbf{W}(t), \mathbf{v}(t))|. \tag{40}$$

This yields

$$\frac{d}{dt} \psi_j(t) \geq \left(1 - \psi_j^2(t)\right) \|\mathbf{x}_+\| - 2n\mathbf{x}_{\max} \max_i |f(\mathbf{x}_i; \mathbf{W}(t), \mathbf{v}(t))| \tag{41}$$

$$\geq \left(1 - \psi_j^2(t)\right) \|\mathbf{x}_+\| - \nu\alpha\|\mathbf{x}_+\|, \tag{42}$$

$$= \|\mathbf{x}_+\| \left(1 - \nu\alpha - \psi_j^2(t)\right), \tag{43}$$

where Equation (42) is due to Lemma D.3. Now, we consider two cases:

- **Case 1.** We have $\psi_j(t) < \sqrt{1 - \nu\alpha}$ for all $t \in [t_1, t_\alpha]$.

- **Case 2.** There exists some $t \in [t_1, t_\alpha]$ such that $\psi_j(t) \geq \sqrt{1 - \nu\alpha}$.

For **case 1**, Lemma D.2 implies that

$$\|\mathbf{x}_+\|\sqrt{1 - \nu\alpha} \leq \frac{d}{dt} \operatorname{arctanh}\left(\frac{\psi_j(t)}{\sqrt{1 - \nu\alpha}}\right). \tag{44}$$

Integrating both sides over $[t_1, t_\alpha]$, we get

$$(t_\alpha - t_1)\|\mathbf{x}_+\|\sqrt{1 - \nu\alpha} \leq \operatorname{arctanh}\left(\frac{\psi_j(t_\alpha)}{\sqrt{1 - \nu\alpha}}\right) - \operatorname{arctanh}\left(\frac{\psi_j(t_1)}{\sqrt{1 - \nu\alpha}}\right) \tag{45}$$

$$\leq \operatorname{arctanh}\left(\frac{\psi_j(t)}{\sqrt{1 - \nu\alpha}}\right) \tag{46}$$

where the second inequality holds as $\psi_j(t_1)$ is nonnegative. Taking $\tanh(\cdot)$ on both sides and scaling, we get the claim.

For **case 2**, we cannot directly apply Lemma D.2, as the $\operatorname{arctanh}(x)$ is undefined for $x \geq 1$. Instead we consider

$$\widetilde{\psi_j}(t) := \min\left\{\psi_j(t), \sqrt{1 - \nu\alpha}\right\}. \tag{47}$$

Then, we know that the differential inequality in Lemma D.2 holds almost everywhere. Indeed, if $\psi_j(t) < \sqrt{1 - \nu\alpha}$, then $\widetilde{\psi_j}(t) = \psi_j(t)$; otherwise, both the LHS and the RHS equals to zero. Noticing $\widetilde{\psi_j}(t) \leq \psi_j(t)$, we get the first claim.

Moreover, sign preservation (Lemma D.1) induced by Assumption 4.1, we have $v_j(t_\alpha) \geq 0$ for all $j \in V_+$. Since the effective predictor lies in the conical hull of $\mathbf{W}(t_\alpha)$, a property that holds independent of the specific non-negative values of $v_j(t_\alpha)$. Thus, we obtain the same lower bound for the effective predictor.

## C.2. Proof of Corollary 5.3

We can proceed as:

$$\sin^2(\angle(\mathbf{w}_j(t_\alpha), \mathbf{x}_+)) = 1 - \psi_j(t_\alpha)^2 \tag{48}$$

$$\leq 1 - (1 - \nu\alpha) \tanh^2 \left((t_\alpha - t_1)\|\mathbf{x}_+\|\sqrt{1 - \nu\alpha}\right) \tag{49}$$

$$= \nu\alpha + (1 - \nu\alpha)\left(1 - \tanh^2 \left((t_\alpha - t_1)\|\mathbf{x}_+\|\sqrt{1 - \nu\alpha}\right)\right) \tag{50}$$

$$= \nu\alpha + (1 - \nu\alpha) \operatorname{sech}^2 \left((t_\alpha - t_1)\|\mathbf{x}_+\|\sqrt{1 - \nu\alpha}\right) \tag{51}$$

$$\leq \nu\alpha + 4(1 - \nu\alpha)\exp\left(-2(t_\alpha - t_1)\|\mathbf{x}_+\|\sqrt{1 - \nu\alpha}\right) \tag{52}$$

$$= \nu\alpha + 4(1 - \nu\alpha)\exp\left(2t_1\|\mathbf{x}_+\|\sqrt{1 - \nu\alpha}\right)\exp\left(-2t_\alpha\|\mathbf{x}_+\|\sqrt{1 - \nu\alpha}\right) \tag{53}$$

$$= \nu\alpha + 4(1 - \nu\alpha)\exp\left(2t_1\|\mathbf{x}_+\|\sqrt{1 - \nu\alpha}\right) \cdot (h\alpha)^{\frac{\|\mathbf{x}_+\|\sqrt{1-\nu\alpha}}{4n\mathbf{x}_{\max}}}. \tag{54}$$

Here, the first inequality follows from Lemma 5.2, the second inequality follows from the fact that $\operatorname{sech}^2(x) \leq 4\exp(-2x)$. Thus, we have

$$\angle(\alpha) \leq \arcsin\left(\sqrt{\nu\alpha + 4(1 - \nu\alpha)\exp\left(2t_1\|\mathbf{x}_+\|\sqrt{1 - \nu\alpha}\right) \cdot (h\alpha)^{\frac{\|\mathbf{x}_+\|\sqrt{1-\nu\alpha}}{4n\mathbf{x}_{\max}}}}\right) \tag{55}$$

$$\leq \frac{\pi}{2}\sqrt{\nu\alpha} + \pi\sqrt{1 - \nu\alpha}\exp\left(t_1\|\mathbf{x}_+\|\sqrt{1 - \nu\alpha}\right) \cdot (h\alpha)^{\frac{\|\mathbf{x}_+\|\sqrt{1-\nu\alpha}}{8n\mathbf{x}_{\max}}} \tag{56}$$

$$\leq C_1\sqrt{\alpha} + C_2\alpha^{\|\mathbf{x}_+\|/8n\mathbf{x}_{\max}} \tag{57}$$

for some $C_1, C_2 > 0$. In terms of $\psi_j(t_\alpha)$ itself, we have $\psi_j(t_\alpha) = 1 - O(\alpha - \alpha^k) \approx 1 - O(\alpha)$, where $k = \|\mathbf{x}_+\|/4n\mathbf{x}_{\max}$.

## C.3. Proof of Lemma 5.4

Since $t \geq t_\alpha$, we have $\psi_j(t) \geq \lambda$. For any $t \geq t_\alpha$, we have

$$\frac{d}{dt}\psi_j(t) \geq \left(\left\langle \frac{\mathbf{x}_+}{\|\mathbf{x}_+\|}, \frac{\sum_{i \in \mathcal{I}_+} c_i(t)\mathbf{x}_i}{\|\sum_{i \in \mathcal{I}_+} c_i(t)\mathbf{x}_i\|}\right\rangle - \psi_j(t)\right)\left\|\sum_{i \in \mathcal{I}_+} c_i(t)\mathbf{x}_i\right\| \tag{58}$$

$$\geq (\lambda - \psi_j(t))\left\|\sum_{i \in \mathcal{I}_+} c_i(t)\mathbf{x}_i\right\|. \tag{59}$$

where $c_i(t) = -\nabla\ell\left(y_i, f(\mathbf{x}_i, \theta_t)\right)$ denotes the loss gradient of $i$th sample at time $t$, and we have used Lemma 5 of Min et al. (2024) for the last inequality. From now on, we write $\mathbf{x}_c(t) = \sum_{i \in \mathcal{I}_+} c_i(t)\mathbf{x}_i$ for notational simplicity.

Now, let $\delta(t) = \psi(t) - \lambda$. Then, multiplying $G(t) := \int_{t_\alpha}^t \|\mathbf{x}_c(\tau)\|d\tau$ for both sides and differentiation gives

$$\frac{d}{dt}(\delta(t) \cdot \exp(G(t)) = \exp(G(t))\frac{d}{dt}\delta(t) + \delta(t)\exp(G(t))\frac{d}{dt}G(t) \tag{60}$$

$$= \exp(G(t))\frac{d}{dt}\delta(t) + \delta(t)\exp(G(t))\|\mathbf{x}_c(t)\| \tag{61}$$

$$\geq 0, \tag{62}$$

where we use Equation (59) for the last inequality. Integrating both sides from $t_\alpha$ to $t_{\eta,\alpha}$, we get

$$\delta(t_{\eta,\alpha})\exp(G(t_{\eta,\alpha})) - \delta(t_\alpha) \geq 0, \tag{63}$$

where we use $\exp(G(t_\alpha)) = 1$. Rewriting the terms, we have

$$\psi(t_{\eta,\alpha}) \geq \lambda + (\psi(t_\alpha) - \lambda)\exp(-G(t_{\eta,\alpha})). \tag{64}$$

Plugging the results of Lemma C.2, we get what we want.

**Lemma C.1.** *We have*

$$\|\mathbf{x}_c(t)\| \leq \mathbf{x}_{\max} n \hat{L}(t). \tag{65}$$

*Proof.* For $y_i = +1$, logistic loss satisfies

$$c_i(t) = -\partial_f \ell(+1, f(\mathbf{x}_i)) = \frac{1}{1 + \exp(f_{\theta_t}(\mathbf{x}_i))} = \frac{u_i(t)}{1 + u_i(t)}. \tag{66}$$

where we let $u_i(t) := \exp(-f_{\theta_t}(\mathbf{x}_i)) \geq 0$. Since $\log(1 + u_i(t)) \geq u_i(t)/(1 + u_i(t))$, we have

$$\frac{d}{du}\left(\log(1 + u_i(t)) - \frac{u_i(t)}{1 + u_i(t)}\right) = \frac{u_i(t)}{(1 + u_i(t))^2} \geq 0, \tag{67}$$

and now we have $c_i(t) \leq \ell(+1, f_{\theta_t}(\mathbf{x}_i))$. Then, we finally have

$$\|\mathbf{x}_c(t)\| = \left\| \sum_{i \in \mathcal{I}_+} c_i(t)\mathbf{x}_i \right\| \tag{68}$$

$$\leq \sum_{i \in \mathcal{I}_+} c_i(t) \|\mathbf{x}_i\| \tag{69}$$

$$\leq \mathbf{x}_{\max} \sum_{i \in \mathcal{I}_+} \ell(+1, f_{\theta_t}(\mathbf{x}_i)) \tag{70}$$

$$= \mathbf{x}_{\max} n \hat{L}(\theta_t), \tag{71}$$

and we get the claim. $\square$

**Lemma C.2.** *We have*

$$G(t_{\eta,\alpha}) \leq \mathbf{x}_{\max} n \left( (t_2 - t_\alpha)\hat{L}(t_\alpha) + \frac{1}{\beta} \log \frac{\hat{L}(t_2)}{\eta} \right), \tag{72}$$

*where* $\beta := (\lambda \mathbf{x}_{\min})^2/(32\mathbf{x}_{\max})$.

*Proof.* By the definition of $G(t)$ and the results of Lemma C.1, we have

$$G(t_{\eta,\alpha}) \leq \mathbf{x}_{\max} n \int_{t_\alpha}^{t_{\eta,\alpha}} \hat{L}(t) dt \tag{73}$$

$$= \mathbf{x}_{\max} n \left( \int_{t_\alpha}^{t_2} \hat{L}(t) dt + \int_{t_2}^{t_{\eta,\alpha}} \hat{L}(t) dt \right), \tag{74}$$

$$\leq \mathbf{x}_{\max} n \left( (t_2 - t_\alpha)\hat{L}(t_\alpha) + \int_{t_2}^{t_{\eta,\alpha}} \frac{\hat{L}(t_2)}{1 + \beta \hat{L}(t_2)(t - t_2)} dt \right) \tag{75}$$

$$= \mathbf{x}_{\max} n \left( (t_2 - t_\alpha)\hat{L}(t_\alpha) + \frac{1}{\beta} \log\left(1 + \beta \hat{L}(t_2)(t_{\eta,\alpha} - t_2)\right) \right) \tag{76}$$

$$\leq \mathbf{x}_{\max} n \left( (t_2 - t_\alpha)\hat{L}(t_\alpha) + \frac{1}{\beta} \log \frac{\hat{L}(t_2)}{\eta} \right). \tag{77}$$

Here, Equation (75) is due to monotonic decreasing property of the training risk and the results from Min et al. (2024, Appendix D.2), and $\beta := (\lambda \mathbf{x}_{\min})^2/(32\mathbf{x}_{\max})$. $\square$

## C.4. Proof of Theorem 5.6

For the proof, we will derive the upper bounds of the each term, $\mathsf{OA}(\alpha)$ and $\mathsf{OF}(\alpha)$, each, and then combine the terms. For notational simplicity, we will write $\bar{\mathbf{x}}_+ = \mathbf{x}_+/\|\mathbf{x}_+\|$.

**Exact formula for the over-alignment, $\mathsf{OA}(\alpha)$.** Thanks to the closed-form expression of the zero-one error under the Gaussian mixture (Lemma D.4), we have

$$\inf_{\mathbf{v} \in H(\alpha)} \mathcal{E}(\mathbf{v}) = \Phi\left(-\sigma^{-1}\mathbf{v}_*^\top \mathbf{s}_+\right), \quad \text{where} \quad \mathbf{v}_* = \arg\max_{\mathbf{v} \in H(\alpha)} \mathbf{v}^\top \mathbf{s}_+. \tag{78}$$

Moreover, $\mathcal{E}^*$ denotes the Bayes error, i.e.,

$$\mathcal{E}^* = \inf_{\mathbf{v} \in \mathbb{R}^d} \mathcal{E}(\mathbf{v}) = \Phi\left(-\sigma^{-1}\right). \tag{79}$$

Combining these terms, we have

$$\mathsf{OA}(\alpha) = \Phi\left(-\mathbf{v}_*^\top \mathbf{s}_+ \sigma^{-1}\right) - \Phi\left(-\sigma^{-1}\right), \tag{80}$$

and we get what we want.

**Bounds for the over-fitting.** For the overfitting term, we derive upper bounds based on both Rademacher and Gaussian complexities, and take their minimum to obtain a tighter bound.

By the scale invariance of the zero-one error (Lemma D.4), we have

$$\mathsf{OF}(\alpha) = \mathcal{E}(\hat{\mathbf{w}}_\alpha) - \inf_{\mathbf{v} \in H(\alpha)} \mathcal{E}(\mathbf{v}) \tag{81}$$

$$= \mathcal{E}(\bar{\mathbf{w}}_\alpha) - \inf_{\mathbf{v} \in H(\alpha)} \mathcal{E}(\mathbf{v}) \tag{82}$$

$$= \Phi\left(-\sigma^{-1}\bar{\mathbf{w}}_\alpha^\top \mathbf{s}_+\right) - \inf_{\mathbf{v} \in H(\alpha)} \Phi\left(-\sigma^{-1}\mathbf{v}^\top \mathbf{s}_+\right) \tag{83}$$

$$= \Phi\left(-\sigma^{-1}\bar{\mathbf{w}}_\alpha^\top \mathbf{s}_+\right) - \Phi\left(-\sigma^{-1}\sup_{\mathbf{v} \in H(\alpha)} \mathbf{v}^\top \mathbf{s}_+\right), \tag{84}$$

where $\bar{\mathbf{w}}_\alpha := \hat{\mathbf{w}}_\alpha/\|\hat{\mathbf{w}}_\alpha\|$. Since $\Phi'(t) = \exp(-t^2/2)/\sqrt{2\pi} \leq 1/\sqrt{2\pi}$, the mean value theorem implies that there exists some $z \in \left[-\sigma^{-1}\sup_{\mathbf{v} \in H(\alpha)} \mathbf{v}^\top \mathbf{s}_+, -\sigma^{-1}\bar{\mathbf{w}}_\alpha^\top \mathbf{s}_+\right]$ such that

$$\Phi\left(-\sigma^{-1}\bar{\mathbf{w}}_\alpha^\top \mathbf{s}_+\right) - \Phi\left(-\sigma^{-1}\sup_{\mathbf{v} \in H(\alpha)} \mathbf{v}^\top \mathbf{s}_+\right) = \Phi'(z)\left(-\sigma^{-1}\bar{\mathbf{w}}_\alpha^\top \mathbf{s}_+ + \sigma^{-1}\sup_{\mathbf{v} \in H(\alpha)} \mathbf{v}^\top \mathbf{s}_+\right) \tag{85}$$

$$\leq \frac{1}{\sqrt{2\pi}}\left(\frac{\sup_{\mathbf{v} \in H(\alpha)} \mathbf{v}^\top \mathbf{s}_+ - \bar{\mathbf{w}}_\alpha^\top \mathbf{s}_+}{\sigma}\right). \tag{86}$$

By Lemma D.5, we have $L(\bar{\mathbf{w}}_\alpha) = \mathbb{E}_{G \sim \mathcal{N}(0,1)}[\ell(\bar{\mathbf{w}}_\alpha^\top \mathbf{s}_+ + \sigma G)]$. Moreover, by the Cauchy-Schwarz inequality, we have $\bar{\mathbf{w}}_\alpha^\top \mathbf{s}_+ \in [-\|\mathbf{s}_+\|, \|\mathbf{s}_+\|]$.

Now define the scalar function $\tilde{L}$ by $\tilde{L}(t) := L(\bar{\mathbf{w}}_\alpha)$, where $t := \bar{\mathbf{w}}_\alpha^\top \mathbf{s}_+$. Note that the logistic loss function $\ell(u)$ is monotonically decreasing with respect to $u \in \mathbb{R}$. Then, since $t \leq \|\mathbf{s}_+\|$, we obtain

$$\tilde{L}'(t) = -\mathbb{E}_{G \sim \mathcal{N}(0,1)}\left[\frac{1}{1 + \exp(t + \sigma G)}\right] \tag{87}$$

$$\leq -\mathbb{E}_{G \sim \mathcal{N}(0,1)}\left[\frac{1}{1 + \exp(\|\mathbf{s}_+\| + \sigma G)}\right] \tag{88}$$

$$\leq -\mathbb{E}_{G \sim \mathcal{N}(0,1)}\left[\frac{1}{1 + \exp(\|\mathbf{s}_+\| + \sigma G)}\mathbb{1}[G \leq 0]\right] \tag{89}$$

$$\leq -\Pr(G \leq 0) \cdot \frac{1}{1 + \exp(\|\mathbf{s}_+\|)} \tag{90}$$

$$= -\frac{1}{2(1 + \exp(\|\mathbf{s}_+\|))}. \tag{91}$$

Here, Equation (90) we use monotonicity of $1/(1 + e^t)$.

Using mean value theorem again for $\tilde{L}(t)$, we have

$$\tilde{L}\left(\sup_{\mathbf{v} \in H(\alpha)} \mathbf{v}^\top \mathbf{s}_+\right) - \tilde{L}\left(\bar{\mathbf{w}}_\alpha^\top \mathbf{s}_+\right) \leq -\frac{1}{2(1 + \exp(\|\mathbf{s}_+\|))}\left(\sup_{\mathbf{v} \in H(\alpha)} \mathbf{v}^\top \mathbf{s}_+ - \bar{\mathbf{w}}_\alpha^\top \mathbf{s}_+\right), \tag{92}$$

which directly implies

$$\sup_{\mathbf{v} \in H(\alpha)} \mathbf{v}^\top \mathbf{s}_+ - \bar{\mathbf{w}}_\alpha^\top \mathbf{s}_+ \leq 2(1 + \exp(\|\mathbf{s}_+\|))\left(\tilde{L}\left(\bar{\mathbf{w}}_\alpha^\top \mathbf{s}_+\right) - \tilde{L}\left(\sup_{\mathbf{v} \in H(\alpha)} \mathbf{v}^\top \mathbf{s}_+\right)\right). \tag{93}$$

Combining Equation (86) and Equation (93), we proceed as

$$\mathsf{OF}(\alpha) \leq \frac{2(1 + \exp(\|\mathbf{s}_+\|))}{\sigma\sqrt{2\pi}}\left(\tilde{L}\left(\bar{\mathbf{w}}_\alpha^\top \mathbf{s}_+\right) - \tilde{L}\left(\sup_{\mathbf{v} \in H(\alpha)} \mathbf{v}^\top \mathbf{s}_+\right)\right) \tag{94}$$

$$= \frac{2(1 + \exp(\|\mathbf{s}_+\|))}{\sigma\sqrt{2\pi}}\left(\tilde{L}\left(\bar{\mathbf{w}}_\alpha^\top \mathbf{s}_+\right) - \inf_{\mathbf{v} \in H(\alpha)} \tilde{L}\left(\mathbf{v}^\top \mathbf{s}_+\right)\right) \tag{95}$$

$$= \frac{2(1 + \exp(\|\mathbf{s}_+\|))}{\sigma\sqrt{2\pi}}\left(L\left(\bar{\mathbf{w}}_\alpha\right) - \inf_{\mathbf{v} \in H(\alpha)} L\left(\mathbf{v}\right)\right) \tag{96}$$

$$\leq \frac{2(1 + \exp(\|\mathbf{s}_+\|))}{\sigma\sqrt{2\pi}}\left(2\sup_{\mathbf{v} \in H(\alpha)} \left|L(\mathbf{v}) - \hat{L}(\mathbf{v})\right| + \hat{L}(\bar{\mathbf{w}}_\alpha) - \inf_{\mathbf{v} \in H(\alpha)} \hat{L}(\mathbf{v})\right) \tag{97}$$

$$\leq \frac{2(1 + \exp(\|\mathbf{s}_+\|))}{\sigma\sqrt{2\pi}}\left(2\sup_{\mathbf{v} \in H(\alpha)} \left|L(\mathbf{v}) - \hat{L}(\mathbf{v})\right| + \eta\right). \tag{98}$$

In Equation (97), we use classical error decomposition technique. Moreover, Equation (98) is due to:

$$\hat{L}(\bar{\mathbf{w}}_\alpha) - \inf_{\mathbf{v} \in H(\alpha)} \hat{L}(\mathbf{v}) \leq \hat{L}\left(\hat{\mathbf{w}}_\alpha / \|\hat{\mathbf{w}}_\alpha\|\right) \leq \hat{L}\left(\hat{\mathbf{w}}_\alpha\right) = \eta, \tag{99}$$

where the inequalities are from $\|\hat{\mathbf{w}}_\alpha\| \leq 1$ and $\langle \hat{\mathbf{w}}_\alpha, \mathbf{x}_i \rangle \geq 0$ for all $i \in \mathcal{I}_+$.

We now proceed to derive two different upper bounds for the uniform deviation, based on Rademacher and Gaussian complexities (Bartlett & Mendelson, 2002), and take the minimum. To bound this term, it is important to note that the logistic loss is sample-wise unbounded due to the unbounded support of Gaussian mixtures, which makes it tricky to apply classical Rademacher complexity-based bounds involving McDiarmid's inequality (Telgarsky, 2021). Instead, we can utilize the Lipschitz property of the logistic loss (i.e., 1-Lipschitz with respect to margin), which is studied from Maurer & Pontil (2021). Note that, since we are handling only the positive-class data, for simplicity, we can let $\mathbf{a}_i := y_i \mathbf{x}_i$, which follows that $y_i \mathbf{v}^\top \mathbf{x}_i = \mathbf{v}^\top \mathbf{a}_i$ and $\mathbf{a}_i$ is the i.i.d. random variable from $\mathcal{N}(\mathbf{s}_+, \sigma^2 \mathbf{I}_d)$.

First, we define a *data-independent* cone $\bar{H}(r)$

$$\bar{H}(r) := \{\mathbf{v} \in \mathbb{S}^{d-1} : \langle \mathbf{s}_+, \mathbf{v} \rangle \geq r\} \tag{100}$$

and the corresponding function class $\bar{\mathcal{H}}_r$

$$\bar{\mathcal{H}}_r := \{\mathbf{a} \mapsto \ell(\mathbf{v}^\top \mathbf{a}) : \mathbf{v} \in \bar{H}(r)\} \tag{101}$$

Moreover, we define a fixed grid $G_\epsilon$ as follows:

$$G_\epsilon := \{-1 + k\epsilon : k = 0, 1, \cdots, \lceil 2/\epsilon \rceil\} \cap [-1, 1]. \tag{102}$$

For the proof, we set $\epsilon \in (0, 0.25)$. For each fixed $r \in G_\epsilon$, we first derive the uniform deviation to $\bar{\mathcal{H}}_r$.

Let i.i.d. random variables $\mathbf{a}_1, \cdots, \mathbf{a}_n \sim \mathcal{N}(\mathbf{s}_+, \sigma^2 \mathbf{I}_d)$ and the empirical covariance matrix $\widehat{\Sigma}_n := \frac{1}{n} \sum_{i=1}^{n} \mathbf{a}_i \mathbf{a}_i^\top$. Then, the upper bound of the (one-side) uniform deviation is

$$\sup_{\mathbf{v} \in \bar{H}(r)} \left( L(\mathbf{v}) - \hat{L}(\mathbf{v}) \right) \le \mathbb{E}_\mathbf{a}[R(\bar{\mathcal{H}}_r, (\mathbf{a}_1, \mathbf{a}_2, \cdots, \mathbf{a}_n))] + 16e \cdot \text{Lip}(\ell) \cdot \|\|\mathbf{a}_1\|\|_{\psi_1} \sqrt{\frac{\log(3/\delta\epsilon)}{n}} \tag{103}$$

$$\le \mathbb{E}_\mathbf{a} \mathbb{E}_\epsilon \left[ \frac{2}{n} \sup_{\mathbf{v} \in \bar{H}(r)} \left( \sum_{i=1}^{n} \epsilon_i \ell(\mathbf{v}^\top \mathbf{a}_i) \right) \right] + 16e \|\|\mathbf{a}_1\|\|_{\psi_1} \sqrt{\frac{\log(3/\delta\epsilon)}{n}} \tag{104}$$

$$\le \sqrt{\frac{\pi}{2}} \cdot \mathbb{E}_\mathbf{a} \underbrace{\mathbb{E}_{g \sim \mathcal{N}(0,1)} \left[ \frac{2}{n} \sup_{\mathbf{v} \in \bar{H}(r)} \left( \sum_{i=1}^{n} g_i \ell(\mathbf{v}^\top \mathbf{a}_i) \right) \right]}_{=: \hat{\mathfrak{S}}_n(\bar{\mathcal{H}}_r)} + 16e \|\|\mathbf{a}_1\|\|_{\psi_1} \sqrt{\frac{\log(3/\delta\epsilon)}{n}}. \tag{105}$$

$$\le \sqrt{2\pi} \left( 1 + \sigma \left( 1 + \sqrt{\frac{d}{n}} \right) \right) \cdot \sqrt{\frac{d(1 - r^2)}{n}} + C_1(1 + \sigma\sqrt{d}) \sqrt{\frac{\log(3/\delta\epsilon)}{n}}, \tag{106}$$

for some $C_1 > 0$. Here, $R(\cdot, \cdot)$ denotes the empirical Rademacher complexity and $\epsilon_i$ denotes the Rademacher random variable. Note that Equation (105) is from standard comparison inequality between Rademacher and Gaussian complexities (Ledoux & Talagrand, 1991).

Taking a union bound for (1) $\bar{\mathcal{H}}_r$ and $-\bar{\mathcal{H}}_r$ and (2) over all $r \in G_\epsilon$, we obtain, with probability at least $1 - \delta$,

$$\sup_{\mathbf{v} \in \bar{H}(r)} \left| L(\mathbf{v}) - \hat{L}(\mathbf{v}) \right| \le \sqrt{2\pi} \left( 1 + \sigma \left( 1 + \sqrt{\frac{d}{n}} \right) \right) \cdot \sqrt{\frac{d(1 - r^2)}{n}} + C_1(1 + \sigma\sqrt{d}) \sqrt{\frac{\log(3/\delta\epsilon)}{n}} \tag{107}$$

Now, after the training samples are realized, define $\tau(\alpha) := \inf_{\mathbf{v} \in H(\alpha)} \langle \mathbf{s}_+, \mathbf{v} \rangle$ and choose

$$r(\alpha) := \max \{ r \in G_\epsilon : 0 < r \le \tau(\alpha) \}. \tag{108}$$

Then, we have $r(\alpha) \in [\tau_\alpha - \epsilon, \tau_\alpha]$. Moreover, since $r(\alpha) \le \tau(\alpha)$, we have

$$H(\alpha) \subseteq \bar{H}(r(\alpha)). \tag{109}$$

Therefore, we obtain

$$\sup_{\mathbf{v} \in H(\alpha)} \left| L(\mathbf{v}) - \hat{L}(\mathbf{v}) \right| \le \sup_{\mathbf{v} \in \bar{H}(r(\alpha))} \left| L(\mathbf{v}) - \hat{L}(\mathbf{v}) \right| \tag{110}$$

$$\le \sqrt{2\pi} \left( 1 + \sigma \left( 1 + \sqrt{\frac{d}{n}} \right) \right) \cdot \sqrt{\frac{d(1 - r^2)}{n}} + C_1(1 + \sigma\sqrt{d}) \sqrt{\frac{\log(6/\delta\epsilon)}{n}}. \tag{111}$$

Plugging Equation (111) into Equation (98), we get the upper bound for $\mathsf{OF}(\alpha)$.

**Combining $\mathsf{OA}$ and $\mathsf{OF}$.** As a final step, combining the derived upper bound for $\mathsf{OF}(\alpha)$ and Equation (80), we get what we want.

C.4.1. DETAILED DERIVATION OF EQUATION (106)

Here, we derive a upper bound for the empirical Gaussian complexity $\hat{\mathfrak{G}}_n\left(\bar{\mathcal{H}}_r\right)$. By defining $\hat{\Sigma}_n := \frac{1}{n}\sum_{i=1}^n \mathbf{a}_i \mathbf{a}_i^\top$, we have

$$\mathbb{E}_{\mathbf{a}}\hat{\mathfrak{G}}_n\left(\bar{\mathcal{H}}_r\right) := \mathbb{E}_{\mathbf{a}}\mathbb{E}_g\left[\frac{2}{n}\sup_{\mathbf{v}\in\bar{H}(r)}\sum_{i=1}^n g_i\ell\left(\mathbf{v}^\top\mathbf{a}_i\right)\,\middle|\,\mathbf{a}_1,\ldots,\mathbf{a}_n\right] \tag{112}$$

$$= \mathbb{E}_{\mathbf{a}}\mathbb{E}_g\left[\frac{2}{n}\sup_{\mathbf{v}\in\bar{H}(r)}\sum_{i=1}^n g_i\left(\ell\left(\mathbf{v}^\top\mathbf{a}_i\right)-\ell(0)\right)\,\middle|\,\mathbf{a}_1,\ldots,\mathbf{a}_n\right] \tag{113}$$

$$\leq \mathbb{E}_{\mathbf{a}}\mathbb{E}_g\left[\frac{2}{n}\sup_{\mathbf{v}\in\bar{H}(r)}\sum_{i=1}^n g_i\mathbf{v}^\top\mathbf{a}_i\,\middle|\,\mathbf{a}_1,\ldots,\mathbf{a}_n\right] \tag{114}$$

$$= \frac{2}{n}\mathbb{E}_{\mathbf{a}}\mathbb{E}_g\left[\sup_{\mathbf{v}\in\bar{H}(r)}\left\langle\mathbf{v},\sum_{i=1}^n g_i\mathbf{a}_i\right\rangle\,\middle|\,\mathbf{a}_1,\ldots,\mathbf{a}_n\right] \tag{115}$$

$$= \frac{2}{\sqrt{n}}\mathbb{E}_{\mathbf{a}}\mathbb{E}_{\mathbf{g}:=[g_1,\cdots,g_n]^\top}\sup_{\mathbf{v}\in\bar{H}(r)}\left\langle\hat{\Sigma}_n^{1/2}\mathbf{g},\mathbf{v}\right\rangle \tag{116}$$

$$\leq \frac{2}{\sqrt{n}}\mathbb{E}_{\mathbf{a}}\left\|\hat{\Sigma}_n\right\|^{1/2}\mathbb{E}_{\mathbf{g}}\sup_{\mathbf{v}\in\bar{H}(r)}\langle\mathbf{g},\mathbf{v}\rangle \tag{117}$$

$$\leq \frac{2}{\sqrt{n}}\mathbb{E}_{\mathbf{a}}\left\|\hat{\Sigma}_n\right\|^{1/2}\sqrt{d(1-r^2)} \tag{118}$$

$$\leq \frac{2}{\sqrt{n}}\left(1+\sigma\left(1+\sqrt{\frac{d}{n}}\right)\right)\sqrt{d(1-r^2)} \tag{119}$$

Here, the detailed derivations are as follows: We use Sudakov-Fernique inequality in Equation (117).

For the Equation (118), we decompose $\mathbf{v}$ as

$$\mathbf{v} = \sqrt{1-\|\mathbf{z}\|^2}\mathbf{s} + \mathbf{z}\quad\text{where}\quad\mathbf{z}\perp\mathbf{s},\quad\|\mathbf{z}\|\leq\sqrt{1-r^2}. \tag{120}$$

In a similar manner, we can write the Gaussian vector $\mathbf{g}$ as

$$\mathbf{g} = \langle\mathbf{g},\mathbf{s}\rangle\mathbf{s} + \mathbf{g}'\quad\text{where}\quad\mathbf{g}'\perp\mathbf{s}. \tag{121}$$

From Equation (121), we have $\langle\mathbf{g},\mathbf{s}\rangle\sim\mathcal{N}(0,1)$ and $\mathbf{g}_\perp\sim\mathcal{N}(0,\mathbf{I}_{d-1})$. Then, we obtain

$$\mathbb{E}_{\mathbf{g}}\sup_{\mathbf{v}\in\bar{H}(r)}\langle\mathbf{g},\mathbf{v}\rangle = \mathbb{E}\sup_{\mathbf{z}\perp\mathbf{s},\|\mathbf{z}\|\leq\sqrt{1-r^2}}\left[\langle\mathbf{g},\mathbf{s}\rangle\left(\sqrt{1-\|\mathbf{z}\|^2}\right)+\langle\mathbf{g}',\mathbf{z}\rangle\right] \tag{122}$$

$$= \mathbb{E}\sup_{\mathbf{z}\perp\mathbf{s},\|\mathbf{z}\|\leq\sqrt{1-r^2}}\left[\langle\mathbf{g},\mathbf{s}\rangle\left(\sqrt{1-\|\mathbf{z}\|^2}-1\right)+\langle\mathbf{g}',\mathbf{z}\rangle\right] \tag{123}$$

$$\leq \sqrt{1-r^2}\sqrt{d-1} + \frac{1}{\sqrt{2\pi}}(1-r) \tag{124}$$

$$\leq \sqrt{1-r^2}\sqrt{d-1} + \frac{1}{\sqrt{2\pi}}\sqrt{1-r^2} \tag{125}$$

$$= \sqrt{1-r^2}\left(\sqrt{d-1}+\frac{1}{\sqrt{2\pi}}\right) \tag{126}$$

$$\leq \sqrt{d(1-r^2)}. \tag{127}$$

It remains to bound the spectral norm of the empirical covariance to dervie Equation (119). Let $\mathbf{A} := [\mathbf{a}_1, \cdots, \mathbf{a}_n]$, then we have $\widehat{\Sigma}_n = \frac{1}{n}\mathbf{A}\mathbf{A}^\top$. We proceed as

$$\mathbb{E}_{\mathbf{a}}\|\widehat{\Sigma}_n\|_2^{1/2} = \frac{1}{\sqrt{n}}\mathbb{E}_{\mathbf{a}}\|\mathbf{A}\|_2 \tag{128}$$

$$= \frac{1}{\sqrt{n}}\mathbb{E}_{\mathbf{a}}\|\mathbf{s}\mathbf{1}_n^\top + \sigma\mathbf{Z}\|_2 \tag{129}$$

$$\leq \frac{1}{\sqrt{n}}\mathbb{E}_{\mathbf{a}}\left[\sqrt{n} + \sigma\|\mathbf{Z}\|_2\right] \tag{130}$$

$$\leq \frac{1}{\sqrt{n}}\mathbb{E}_{\mathbf{a}}\left[\sqrt{n} + \sigma(\sqrt{n} + \sqrt{d})\right] \tag{131}$$

$$= 1 + \sigma\left(1 + \sqrt{d/n}\right). \tag{132}$$

where $\mathbf{Z}$ has each entry as $\mathcal{N}(0, 1)$.

As a final step, we apply Lemma D.7 to bound $\|\|\mathbf{a}_1\|\|_{\psi_1}$, and we get what we want.

## C.5. Proof of Proposition 5.7

For the empirical mean, we have

$$\bar{\mathbf{x}}_+ = \frac{1}{n}\sum_{i=1}^n \mathbf{x}_i = \kappa \mathbf{s}_+ + \sigma \frac{1}{n}\sum_{i=1}^n \mathbf{z}_i. \tag{133}$$

We decompose $\bar{\mathbf{z}} := \frac{1}{n}\sum_{i=1}^n \mathbf{z}_i$ into

$$\bar{\mathbf{z}} = \langle \bar{\mathbf{z}}, \mathbf{s}_+ \rangle \mathbf{s}_+ + \Pi_{\mathbf{s}_+}^\perp \bar{\mathbf{z}}, \tag{134}$$

where $\Pi_{\mathbf{s}}^\perp \bar{\mathbf{z}}$ is component of $\bar{\mathbf{z}}$ orthogonal to $\mathbf{s}_+$. Then, in terms of $\phi = \angle(\bar{\mathbf{x}}_+, \mathbf{s}_+)$, we can write as

$$\tan\phi = \frac{\sigma \|\Pi_{\mathbf{s}_+}^\perp \bar{\mathbf{z}}\|}{\kappa + \sigma \langle \bar{\mathbf{z}}, \mathbf{s}_+ \rangle}. \tag{135}$$

Now, for $\delta \in (0,1)$, let $t = \log(4/\delta)$. Since $\langle \bar{\mathbf{z}}, \mathbf{s}_+ \rangle \sim \mathcal{N}(0, 1/n)$, using standard Gaussian tail bound, we have

$$\Pr\left(|\langle \bar{\mathbf{z}}, \mathbf{s}_+ \rangle| \le \sqrt{2t/n}\right) \ge 1 - \delta/2. \tag{136}$$

Next, by the fact that $n\|\Pi_{\mathbf{s}_+}^\perp \bar{\mathbf{z}}\|^2 \sim \chi_{d-1}^2$, from Laurent & Massart (2000), with probability at least $1 - \delta/2$, we have

$$d - 1 - 2\sqrt{(d-1)t} \le n\|\Pi_{\mathbf{s}_+}^\perp \bar{\mathbf{z}}\|^2 \le d - 1 + 2\sqrt{(d-1)t} + 2t. \tag{137}$$

Taking a union bound, we have, with probability at least $1 - \delta$, Equations (136) and (137) hold. On this event, if $\kappa/\sigma > \sqrt{2t/n}$, we proceed as

$$\frac{\max\left\{0, d - 1 - 2\sqrt{(d-1)t}\right\}}{n\left(\kappa/\sigma + \sqrt{2t/n}\right)^2} \le \tan^2\phi \le \frac{d - 1 + 2\sqrt{(d-1)t} + 2t}{n\left(\kappa/\sigma - \sqrt{2t/n}\right)^2} \tag{138}$$

$$\implies \frac{\sqrt{\gamma_1}}{\gamma_2} \cdot \frac{\max\left\{0, d - 1 - 2\sqrt{(d-1)t}\right\}}{d\left(1 + \sqrt{\frac{2t}{n\gamma_2\sqrt{d\log n}}}\right)^2} \le \tan^2\phi \le \frac{\sqrt{\gamma_1}}{\gamma_2} \cdot \frac{d - 1 + 2\sqrt{(d-1)t} + 2t}{d\left(1 - \sqrt{\frac{2t}{n\gamma_2\sqrt{d\log n}}}\right)^2}, \tag{139}$$

where we use $\frac{d/n}{\kappa^2/\sigma^2} = \frac{\sqrt{\gamma_1}}{\gamma_2}$.

Consider the regime where $d, n \to \infty$ and $\gamma_2 \to \gamma_{2,\infty} \in (0, \infty)$, we have

$$\frac{\max\left\{0, d - 1 - 2\sqrt{(d-1)t}\right\}}{d} \to 1, \qquad \frac{d - 1 + 2\sqrt{(d-1)t} + 2t}{d} \to 1, \qquad \sqrt{\frac{2t}{n\gamma_2\sqrt{d\log n}}} \to 0. \tag{140}$$

Therefore, we have

$$\tan^2\phi \to \frac{\sqrt{\gamma_1}}{\gamma_2}. \tag{141}$$

Now we consider three different regimes introduced in Proposition 5.7.

**1. Data-abundant regime.** If $\gamma_1 \to 0$, then $\tan^2\phi \to 0$, and $\phi \to 0$, with high probability.

**2. Moderate regime.** If $\gamma_1 \to \gamma_{1,\infty} \in (0, \infty)$, then $\tan^2\phi \to \sqrt{\gamma_{1,\infty}}/\gamma_{2,\infty}$ with high probability. Therefore, we obtain $\phi \to \arctan\left(\gamma_{1,\infty}^{0.25}/\gamma_{2,\infty}^{0.5}\right)$.

**3. High-dimensional regime.** If $\gamma_1 \to \infty$, then $\tan^2\phi \to \infty$ with high probability, and hence $\phi \to \pi/2$,

and this completes the proof.

# D. Technical Lemmata and Known Results

**Lemma D.1** (Gradient flow properties). *For any $j \in [h]$ and $t \geq 0$, we have*

- *(Balancedness, from Du et al. (2018).)* $v_j(t)^2 - \|\mathbf{w}_j(t)\|^2 = 0$.
- *(Sign preservation, from Boursier et al. (2022).)* $\mathrm{sign}(v_j(t)) = \mathrm{sign}(v_j(0))$.

*Proof.* See each paper for the proof. □

**Lemma D.2.** *Let $\phi(t)$ be a differentiable function satisfying $\dot{\phi}(t) \geq b\left(c^2 - \phi^2(t)\right)$ for some $b, c > 0$. Then, we have:*

$$\frac{d}{dt} \mathrm{arctanh}\left(\phi(t)/c\right) \geq bc. \tag{142}$$

*Proof.* First, note that the derivative of the $\mathrm{arctanh}(x)$ is $1/(1 - x^2)$. Then, by the chain rule,

$$\frac{d}{dt} \mathrm{arctanh}(\phi(t)/c) \quad = \quad \frac{c^2}{c^2 - \phi^2(t)} \cdot \frac{d}{dt}(\phi(t)/c) \quad = \quad \frac{c}{c^2 - \phi^2(t)}\dot{\phi}(t) \quad \geq \quad bc \tag{143}$$

where the last inequality follows from the assumption. □

**Lemma D.3** (Lemma 3 and 4 of Min et al. (2024)). *Consider (sub)gradient flow optimization as specified in Section 4.2 and the two-layer ReLU network is initialized with scale $\alpha \leq \frac{1}{4\sqrt{h}\mathbf{x}_{\max}\mathbf{W}_{\max}^2}$. For any $t \leq t_\alpha := \frac{1}{4n\mathbf{x}_{\max}} \log \frac{1}{\sqrt{h}\alpha}$ and $i \in \mathcal{I}_+$, we have*

$$\left\| \frac{d}{dt} \frac{\mathbf{w}_j(t)}{\|\mathbf{w}_j(t)\|} - \mathrm{sgn}(\mathbf{v}_j(0)) \left( \mathbf{I}_h - \frac{\mathbf{w}_j(t)\mathbf{w}_j(t)^\top}{\|\mathbf{w}_j(t)\|^2} \right) \left( \sum_{i=1}^n \mathbf{x}_i y_i \sigma'(\langle \mathbf{x}_i, \mathbf{w}_j(t) \rangle) \right) \right\| \leq 2n\mathbf{x}_{\max} \max_i |f(\mathbf{x}_i; \mathbf{W}(t), \mathbf{v}(t))|. \tag{144}$$

*Proof.* See Min et al. (2024) for the proof. □

**Lemma D.4.** *Let $(\mathbf{x}, y)$ be the drawn sample from the data model from Section 4.2, for any unit vector (predictor) $\mathbf{w} \in \mathbb{S}^{d-1}$ with scalar multiplier $c > 0$, we have*

$$\Pr(\operatorname{sgn}(c\mathbf{w}^\top \mathbf{x}) \neq y) = \Phi\left(-\frac{\mathbf{w}^\top \mathbf{s}}{\sigma}\right). \tag{145}$$

*Proof.* An error occurs iff $yc\mathbf{w}^\top \mathbf{x} \leq 0$. By the assumption, we have $\mathbf{x} = y\mathbf{s} + \sigma\mathbf{z}$,

$$yc\mathbf{w}^\top \mathbf{x} = yc\mathbf{w}^\top(y\mathbf{s} + \sigma\mathbf{z}) = c\mathbf{w}^\top \mathbf{s} + \sigma yc\mathbf{w}^\top \mathbf{z}. \tag{146}$$

Since $\mathbf{z} \sim \mathcal{N}(0, \mathbf{I}_d)$ and $\|\mathbf{w}\| = 1$, we have $\mathbf{w}^\top \mathbf{z} \sim \mathcal{N}(0, 1)$. Moreover, $y$ is independent of $\mathbf{z}$ and $y \in \{\pm 1\}$, so $y\mathbf{w}^\top \mathbf{z} \overset{d}{=} \mathbf{w}^\top \mathbf{z}$. Hence we may write

$$\Pr\left(\operatorname{sign}(c\mathbf{w}^\top \mathbf{x}) \neq y\right) = \Pr\left(\mathbf{w}^\top \mathbf{s} + \sigma G \leq 0\right), \qquad G \sim \mathcal{N}(0, 1) \tag{147}$$

$$= \Pr\left(G \leq -\frac{\mathbf{w}^\top \mathbf{s}}{\sigma}\right) \tag{148}$$

$$= \Phi\left(-\frac{\mathbf{w}^\top \mathbf{s}}{\sigma}\right), \tag{149}$$

and this completes the proof. Note that, since we are interested in positive-class data, this can be handled in exactly the same way. $\square$

**Lemma D.5.** *Suppose the data follow the setup specified in Section 4.2. Let $L(\cdot)$ be the population logistic risk and $\ell(\cdot)$ be the logistic loss function. Then, whenever $\|\mathbf{w}\| = 1$, we have*

$$L(\mathbf{w}) = \mathbb{E}_{G \sim \mathcal{N}(0,1)}\left[\ell(\mathbf{w}^\top \mathbf{s} + \sigma G)\right]. \tag{150}$$

*Proof.* We have $y\mathbf{w}^\top \mathbf{x} = \mathbf{w}^\top \mathbf{s} + \sigma\mathbf{w}^\top \mathbf{z}$. Since $\mathbf{z} \sim \mathcal{N}(\mathbf{0}, \mathbf{I}_d)$, we have $\mathbf{w}^\top \mathbf{z} \sim \mathcal{N}(0, 1)$. Substituting the term, we get the claim. $\square$

**Theorem D.6** (Theorem 9 of Maurer & Pontil (2021)). *Let $X = (X_1, \cdots, X_n)$ be i.i.d. random variables with values in a Banach space $(\mathcal{X}, \|\cdot\|)$ and $\mathcal{H} = \{h : \mathcal{X} \to \mathbb{R}\}$ such that $h(\cdot)$ is $L$-Lipschitz for all $\mathbf{x}, y \in \mathcal{X}$ and $h \in \mathcal{H}$. If $n \geq \log(1/\delta)$ then with probability at least $1 - \delta$, we have*

$$\sup_{h \in \mathcal{H}}\left(\frac{1}{n}\sum_{i=1}^{n} h(X_i) - \mathbb{E}[h(X)]\right) \leq \mathbb{E}[R(\mathcal{H}, X)] + 16eL\|\|X_1\|\|_{\psi_1}\sqrt{\frac{\log(1/\delta)}{n}}, \tag{151}$$

*where $R(\mathcal{H}, X) := \mathbb{E}\left[\frac{2}{n}\mathbb{E}\left[\sup_{h \in \mathcal{H}}\sum_i \epsilon_i h(X_i)|X\right]\right]$ denotes the empirical Rademacher complexity and $\|\cdot\|_{\psi_1}$ denotes the sub-exponential norm.*

*Proof.* Check the original paper for the proof. $\square$

**Lemma D.7.** *Let* $\mathbf{x} = \mathbf{s} + \sigma \mathbf{z}$, *where* $\|\mathbf{s}\| = 1$, $\sigma > 0$, *and* $\mathbf{z} \sim \mathcal{N}(\mathbf{0}, \mathbf{I}_d)$. *Then*

$$\| \|\mathbf{x}\| \|_{\psi_1} = O\left(1 + \sigma\sqrt{d}\right). \tag{152}$$

*Proof.* At first, by the triangle inequality of the Euclidean norm, we have $\|\mathbf{x}\| \leq \|\mathbf{s}\| + \sigma\|\mathbf{z}\| = 1 + \sigma\|\mathbf{z}\|$. Then, using the triangle inequality for the sub-exponential norm, we obtain

$$\| \|\mathbf{x}\| \|_{\psi_1} \leq \|1\|_{\psi_1} + \sigma \| \|\mathbf{z}\| \|_{\psi_1} = \frac{1}{\ln 2} + \sigma \| \|\mathbf{z}\| \|_{\psi_1}. \tag{153}$$

Thus, to derive the bound, it suffices to upper bound $\| \|\mathbf{z}\| \|_{\psi_1}$. By Vershynin (2026, Theorem 3.1.1), we have

$$\left\| \|\mathbf{z}\| - \sqrt{d} \right\|_{\psi_2} \leq C \max_i \|\mathbf{z}_i\|_{\psi_2}, \tag{154}$$

for some $C > 0$. Here, $\|\cdot\|_{\psi_2}$ denotes the sub-Gaussian norm. Since $\max_i \|\mathbf{z}_i\|_{\psi_2} = O(1)$ (Vershynin, 2026), it follows that $\left\| \|\mathbf{z}\| - \sqrt{d} \right\|_{\psi_2} = O(1)$. By the inequality: $\| \|\mathbf{z}\| - \sqrt{d}\|_{\psi_1} \lesssim \| \|\mathbf{z}\| - \sqrt{d}\|_{\psi_2}$, we have

$$\left\| \|\mathbf{z}\| - \sqrt{d} \right\|_{\psi_1} = O(1). \tag{155}$$

Using triangle inequality yields

$$\| \|\mathbf{z}\| \|_{\psi_1} \leq \|\sqrt{d}\|_{\psi_1} + \| \|\mathbf{z}\| - \sqrt{d}\|_{\psi_1} = \frac{\sqrt{d}}{\ln 2} + O(1) = O(\sqrt{d}). \tag{156}$$

Combining the terms, we get the claim [5]. $\qquad \square$

---

[5]Note that the bound can be refined to $\Theta(\cdot)$, but we defer doing so for the sake of simplicity.

# E. Supplementary Theoretical Results

In this section, we provide probabilistic bounds for orthogonal separability under the Gaussian mixture model and present additional results that complement the theoretical analyses in Section 5.

## E.1. Proof about Assumption 4.1

**Proposition E.1.** *There exist universal constants $c_1, c_2, c_3 > 0$ such that the following holds: Consider the data model from Section 4.2. If $d \geq c_1 \log n$ and $\kappa^2 \geq c_2 \sigma^2 \sqrt{d \log n}$, then, with probability at least $1 - 4/n^2$, the training dataset is orthogonally separable, i.e., $\lambda > 0$.*

*Proof.* To show the orthogonal separability of the data, it suffices to show $\langle \mathbf{z}_i, \mathbf{z}_j \rangle > 0$, for all $i, j \in [n]$, where $\mathbf{z}_i := y_i \mathbf{x}_i$.

First, let $a := \sqrt{6 \log n}$. Then, for each $i$, we have $\Pr(|\mathbf{s}^\top \zeta_i| > a) \leq 2/n^3$, where $\zeta_i \sim \mathcal{N}(0, I_d)$ are the noise vectors satisfying $\mathbf{z}_i = \kappa \mathbf{s} + \sigma \zeta_i$. Taking a union bound, we obtain

$$\Pr \left( \max_{i \in [n]} |\mathbf{s}^\top \zeta_i| > a \right) \leq 2/n^2. \tag{157}$$

Using 1-Lipschitzness of $\| \cdot \|$ and standard Gaussian concentration inequality yield $\Pr(\|\zeta_i\| > \sqrt{d} + a) \leq 1/n^3$. Again, with union bound

$$\Pr \left( \max_{i \in [n]} \|\zeta_i\| > \sqrt{d} + a \right) \leq 1/n^2. \tag{158}$$

Next, we are interested in pariwise inner product, i.e., $\langle \zeta_i, \zeta_j \rangle$. Using Bernstein's inequality, we have $\Pr \left( |\langle \zeta_i, \zeta_j \rangle| > c_3 (\sqrt{d \log n} + \log n) \right) \leq 2/n^6$. Applying union bound, we get

$$\Pr \left( \max_{i \neq j} |\langle \zeta_i, \zeta_j \rangle| > c_3 (\sqrt{d \log n} + \log n) \right) \leq 1/n^2. \tag{159}$$

Now, we aggregate the results. Let $E$ be the event on which all bounds from above hold simultaneously. Then we have $\Pr(E) \geq 1 - 4/n^2$. By the definition of $\mathbf{z}_i$, we proceed as follows. For some $C > 0$, we have

$$\langle \mathbf{z}_i, \mathbf{z}_j \rangle = \kappa^2 + \kappa \sigma \mathbf{s}^\top (\zeta_i + \zeta_j) + \sigma^2 \langle \zeta_i, \zeta_j \rangle \tag{160}$$

$$\geq \kappa^2 - 2\kappa \sigma \sqrt{6 \log n} - c_3 \sigma^2 (\sqrt{d \log n} + \log n) \tag{161}$$

$$\geq \kappa^2 - C \kappa \sigma \sqrt{\log n} - C \sigma^2 (\sqrt{d \log n} + \log n). \tag{162}$$

Since $d \geq c_1 \log n$, we have $\log n \lesssim \sqrt{d \log n}$. Moreover, using Young's inequality yields the numerator of Equation (162) to be positive:

$$\kappa^2 \geq c_2 \sigma^2 \sqrt{d \log n}, \tag{163}$$

for some $c_2 > 0$. This concludes the proof. □

**Remark E.2.** Proposition E.1 suggests that, under the data model in Section 4.2, if the data dimension $d$ and the signal strength $\kappa$ is sufficiently large relative to the noise level $\sigma$, then with high probability, orthogonal separability satisfied.

## E.2. Experiments on Phase 2 Dynamics

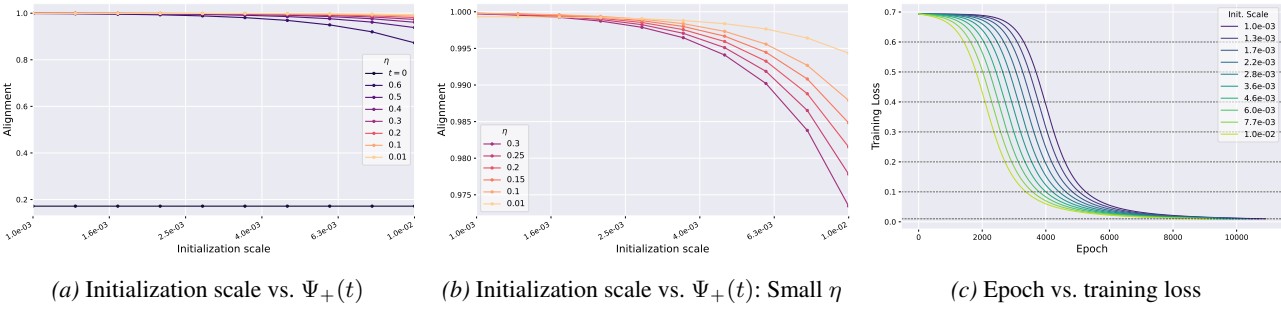

*(a)* Initialization scale vs. $\Psi_+(t)$    *(b)* Initialization scale vs. $\Psi_+(t)$: Small $\eta$    *(c)* Epoch vs. training loss

*Figure 19.* **Phase 2 results.**

Following Section 5.2, this section presents the alignment results obtained in Phase 2. For completeness, we recall Lemma 5.4:

**Lemma E.3** (Lemma 5.4 in the paper)**.** *Let* $\beta := (\lambda \mathbf{x}_{\min})^2/(32\mathbf{x}_{\max})$ *and* $t_2 = O(\log(1/\alpha)/n)$*. Then, for any risk threshold* $\eta > 0$*, we have*

$$\psi_j(t_{\eta,\alpha}) \geq \lambda + m(\alpha)\exp(-g(\alpha)), \tag{164}$$

*where* $m(\alpha) := \psi_j(t_\alpha) - \lambda$ *and* $g(\alpha) \leq \mathbf{x}_{\max}n_+\left((t_2 - t_\alpha)\hat{L}(t_\alpha) + \frac{1}{\beta}\log\frac{\hat{L}(t_2)}{\eta}\right)$.

From the results of Phase 1, we have that $m(\alpha)$ increases as $\alpha$ decreases. Thus, our main interest is in the term $g(\alpha)$. By the definition of $t_2$, which the time that loss significantly decrease, we regard $t_2 \approx t_\alpha$, hence $g(\alpha) \leq \mathbf{x}_{\max}n_+(\log(\hat{L}(t_2)/\eta))/\beta \approx O(1)$, with respect to $\alpha$, thus, we have, in a approximate sense,

$$\psi_j(t_{\eta,\alpha}) \approx \psi_j(t_\alpha). \tag{165}$$

**Interpretation.** Equation (165) suggests that after the alignment phase (Phase 1), the alignment changes very little. To validate this, we examine $\Psi(t_{\eta,\alpha})$—the value of $\Psi(t)$ when each network (for various initialization scales) reaches a target loss level $\eta$. The results are shown in Figure 19a. As can be seen, across all initialization scales, $\Psi$ is already large by the time the loss starts to decrease (see also Figure 19c). More precisely, although all initialization scales start from nearly the same alignment value at $t = 0$, the alignment jumps to a large value immediately after the alignment phase (i.e., at $\eta = 0.6$) and then remains at similar values thereafter. We also observe that $\Psi(t)$ decreases sub-linearly with respect to $\alpha$, which is consistent with Corollary 5.3. This tendency persists even as the target $\eta$ decreases (see Figure 19b), indicating that Equation (165) holds approximately.

**Details of Figure 19.** For training, we use 300 training samples generated from a Gaussian mixture model with $\kappa = 2$, $\sigma = 1$, and $\lambda = 0$. We use two-layer, bias-free ReLU networks with 64 hidden units, and train all models until the training risk reaches $\eta = 0.01$.

