# OpenReview forum: "Over-Alignment vs Over-Fitting: The Role of Feature Learning Strength in Generalization"
_ICML.cc/2026/Conference — ICML 2026 regular_

### Official Review · Reviewer_uWmC · 2026-03-08

**Soundness:** 2
**Presentation:** 3
**Significance:** 2
**Originality:** 3
**Overall Recommendation:** 4
**Confidence:** 3

**Summary:**

This paper studies how the feature learning strength (FLS) affect generalization error. There are two main results: (1) the authors conducted experiments on practical models, and empirically shows that the existence of a "sweet spot" of feature learning strength that minize the test loss. (2) The authors proved that, for two-layer relu network trained on Gaussian mixture model, there is a way to decompose the upper bound of the test error, so that one term is increasing with FLS and the other is decreasing, which indicates the tradeoff of FLS.

**Compliance With Llm Reviewing Policy:**

Affirmed.

**Final Justification:**

This paper studied how feature learning strength affect generalization. I believe it is an interesting problem.

The paper provide proper experimental results to illustrate the phenomenon that the , and studied a case theoretically to understand the insight of the phenomenon. I believe this is a solid contribution, and I believe the paper is above the acceptance line of the conference.

There's still some limitations, such as the theoretical result is on a very specific setting, and proof techniques are not that novel. I do not consider those points are major weaknesses of the paper given the interesting theoretical insights provided, but I also respect that other reviews or AC would place different values on those points.

**Key Questions For Authors:**

- Please also refer to the strengths and weaknesses part.

- I wonder if it is possible to characterize how the scaling of $\alpha$ w.r.t model with affect the test error? In particular, if $\alpha = h^{-\gamma},$ can we find an optimal $\gamma_*$ that lead to the best test error?

**Limitations:**

Yes.

**Strengths And Weaknesses:**

### Strengths

1. This paper studies the problem of how FLS affect generalization error, which is an interesting topic in my opinion.

2. The theoretical parts provide an good example on why large FLS could potentially harm generalization, and the intuition is clean. In particular, if I understand correctly, too large FLS will lead the network weights to over-align with the empirical means, which is in general  not optimal.

### Weaknesses

1. While I believe the theoretical results provide the correct intuition, the main Theorem 5.6 seems to be not strong enough, and a bit hard to interpret. First of all, it only gives an upper bound on the test error; secondly, although the upper bound can be decomposed into two parts that indicates the tradeoff, the discussion on how exactly the tradeoff perform is quite qualitative. While I believe the theoretical results gives the correct tradeoff of test error, I can not directly tell from the formula in Theorem 5.6.  Thus, I wonder:

    (1) Is it possible to find an example of $alpha_1 > \alpha_2,$ so that the $OA(\alpha_1) < OA(\alpha_2),  OF(\alpha_1) > OF(\alpha_2),  OA+OF(\alpha_1) < OA+OF(\alpha_2)$ ?

    (2) Is it possible to characterize the optimal \alpha for the the upper bounds in any case? Could we verify that the optimal $\alpha$ comply with the condition (16)?

    (3) Is it possible to show an lower bound in any case? For example, is it possible to find an example of $alpha_1 > \alpha_2,$ so that $testerror(\alpha_2) > OA(\alpha_1) + OF(\alpha_1)$?

2. The theoretical results consider regime in equation (16), $\alpha < \frac{1}{4 \sqrt{h} x_{max}  W_{\max}^2},$ which is already a regime that strong feature learning happens. It is not clear to me whether there will still be tradeoff outside of this regime.

3. A minor point: it would be beneficial to distinguish the feature learning strength which corresponds to the scaling of the output studied in this paper, and other definition of feature learning, e.g. (Yang & Hu 2021). While the concepts are related, I think the definition is not the same, and it could be a bit confusing for readers that are not familiar with the literature of feature learning strength.

---

> ### Author Rebuttal · Authors · 2026-03-31
>
> Dear reviewer uWMC,
>
> Thank you for your insightful comments.
>
> ---
>
> **Clarifications on W1-(1,2,3)**
>
> Before we begin, based on your response, we believe there may be a few misunderstandings. We clarify them as follows.
>
> - The **OA term in Thm 5.6 is NOT an upper bound---it’s an exact equality**. Whereas for the **OF term, an exact equality cannot be obtained, so it is stated as an upper bound**.
> - In Fig. 4, the OA and OF curves are the actual values computed from Eq. (24). The function $g(\alpha)$ (green line) is included as a proxy to illustrate that the upper bound of OF increases with $\alpha$. Since OA is given by an exact equality, there is no particular need to plot a separate proxy for it.
>
> Then, your W1-(1,2,3) can be resolved as follows (For additional intuition on the OA-OF trade-off, we kindly refer the reviewer to our responses to reviewer xv7o).
>
> ---
>
> **W1-(1). Example of $\alpha_1 > \alpha_2$, so that the $\mathsf{OA}(\alpha_1) < \mathsf{OA}(\alpha_2), \mathsf{OF}(\alpha_1) > \mathsf{OF}(\alpha_2), \mathsf{OA}+\mathsf{OF}(\alpha_1) < \mathsf{OA}+\mathsf{OF}(\alpha_2)$**
>
> This is exactly what Fig. 4 is about: $\mathsf{OA}$ decreases w.r.t. $\alpha$ and $\mathsf{OF}$ increases w.r.t. $\alpha$, and the relation
> $\alpha_1 > \alpha_2 \Rightarrow \mathsf{OA}(\alpha_1) < \mathsf{OA}(\alpha_2), \mathsf{OF}(\alpha_1) >\mathsf{OF}(\alpha_2)$ trivially holds.
>
> In this case, the sum $\mathsf{OA}+\mathsf{OF}$ exhibits the tendency shown in Fig. 14. Roughly speaking, in the regime $\alpha_2 < \alpha_1 < 2e-2$, we have
> $\mathsf{OA}(\alpha_1)+\mathsf{OF}(\alpha_1) < \mathsf{OA}(\alpha_2)+\mathsf{OF}(\alpha_2)$.
>
>
>
>
> ---
>
> **W1-(2). Upper bound of optimal $\alpha$**
>
> That’s a great question.
>
> Differentiating the RHS of Thm 5.6 yields optimal FLS, which is $\alpha^\star \propto \frac{1}{n\sqrt{h}}$, where we have abstracted out the data-dependent terms. Moreover, this approximate optimum is consistent with the upper bound in Eq. (16).
>
> In our simulations (Fig. 14), we also observed that the optimal $\alpha$ satisfies Eq. (16) numerically: Here, the condition for $\alpha$ is approximately $\leq 3e-02$, and the optimal $\alpha$ is below this.
>
> We will add these details in the final version.
>
> ---
>
> **W1-(3). Lower bound in any case**
>
> Yes---As stated in our answer to **W1-(1)**, Fig. 14 shows that when $\alpha_2 < \alpha_1 < 2e-02$, we have $\mathrm{Testerror}(\alpha_2) -\mathcal{E}^{\star}=\mathsf{OA}(\alpha_2)+\mathsf{OF}(\alpha_2) >\mathsf{OA}(\alpha_1)+\mathsf{OF}(\alpha_1)$.
>
> As a remark, we note that RC-based approaches do not provide an analytic lower bound.
>
> ---
>
> **W2. Tradeoff outside of strong feature learning regime**
>
> We first clarify that our work deliberately focuses on the large FLS regime (Eq. (16)), which is more practically relevant and is where the “sweet spot” emerges.
>
> Outside this regime, experimental results suggest that our analysis can still be extrapolated: as $\alpha$ increases, OA vanishes and OF diverges. In fact, many previous works have already analyzed the low FLS regime (a.k.a. kernel regime) as a standalone setting (e.g., [1,2]).
>
> ---
>
> **W3. Distinguish the FLS and other definitions of feature learning**
>
> Thank you for the suggestion. We will add clarifications in the revised version.
>
> ---
>
> **Q1. Optimal FLS vs. width**
>
> As mentioned in **W1-(2)**, extending our theoretical results suggests that the optimal $\alpha$ (i.e., $\alpha^\star$) scales as $\alpha^\star \propto h^{-1/2}$ (or the optimal output multiplier $c^\star \propto h^{-1}$), where $h$ denotes the width.
>
> Our experimental results are close to what our theory predicts. In particular, we trained a 5-layer vanilla CNN with varying widths on a BigGAN-generated dataset. The results show a clear trend in the optimal FLS as a function of width, which aligns with our theoretical analysis.
>
> As a bonus, we provide results on the optimal FLS versus the number of training data (width=$2^6$), derived as $c^\star \propto n^{-2}$, which is also close to our findings.
>
> |Width|$2^4$|$2^6$|$2^8$|
> |-|-|-|-|
> | $c^\star$|$2^{-4}$|$2^{-6}$|$2^{-8}$|
>
> |Training set size|$\times1.0$|$\times0.5$|$\times0.1$|
> |-|-|-|-|
> | $c^\star$|$2^{-6}$|$2^{-5}$|$2^{-4}$|
>
> For a more precise scaling argument, we leave it as future work, as discussed in sec. 6.
>
> ---
>
> [1] Arora et al., Fine-Grained Analysis of Optimization and Generalization…, ICML’19
>
> [2] Cao & Gu, Generalization Bounds of Stochastic Gradient Descent…, NeurIPS’19

---

> > ### Author Rebuttal · Reviewer_uWmC · 2026-04-04
> >
> > I thank the reviewer for the detailed reply to my questions and concerns.
> >
> > In the original review, my major concern is on the strength of the the main theory. In the rebuttal, the author address my major concerns by discussing more theoretical details. Thus I'm willing to increase my score to 4.

---

> > > ### Author Response · Authors · 2026-04-04
> > >
> > > Dear reviewer uWmC,
> > >
> > > We are grateful to hear that our responses addressed your concerns, and we also sincerely appreciate your decision to raise the score.
> > >
> > > Best regards,
> > > Authors

---

### Official Review · Reviewer_GqSR · 2026-03-11

**Soundness:** 2
**Presentation:** 3
**Significance:** 2
**Originality:** 3
**Overall Recommendation:** 4
**Confidence:** 4

**Summary:**

This work studies Feature learning strength (FLS) which plays a critical role in shaping the optimization dynamics of neural nets (NNs). FLS is defined as the inverse of the effective scaling applied to the model output, which can be controlled by the initialization scale or an explicit output multiplier. When FLS is large, features evolve nonlinearly throughout training, reflecting genuine feature learning, when FLS is small, training closely resembles kernel learning, with features remaining largely fixed.

A substantial body of prior works has shown that analyzing these two regimes — feature/rich learning vs. kernel/lazy learning — yields valuable insights into the optimization dynamics and generalization of deep learning. However, theoretical understanding of how FLS affects generalization is poorly aligned with practical observations, i.e., in finite-time training. This manuscript addresses the question whether stronger feature learning improves generalization? And if not, why?

Empirical work reported in this paper uncovers the emergence of an optimal mid-way FLS that yields substantial generalization gains. This is done for image classification tasks using the VGG and ResNet architectures. To illuminate their empirical results, the authors develop a theory for gradient flow dynamics in two-layer ReLU NNs trained with logistic loss, where FLS is controlled via weight scaling at initialization. In this setting they analyze the optimization dynamics induced by varying the FLS.

The authors rely on work (Min et al., 2024; Boursier & Flammarion, 2025) to study gradient flow (GF) in the feature learning regime. They first show that the initialization scale governs the angular deviation of the weights. They then derive an error bound decomposed into an over-alignment term and an over-fitting term (Theorem 5.6). That’s their main theoretical result which establishes the existence of an optimal FLS that interpolates between over-alignment (large FLS) and over-fitting (small FLS).

**Compliance With Llm Reviewing Policy:**

Affirmed.

**Key Questions For Authors:**

* I wonder about a training time dimension of Figure 2. In practical terms, could we be paying significantly in training budget to get just a bit of extra generalization performance? Since we’re changing the learning rate and wait for 99% train Acc, training time may vary significantly. Is it feasible to consider hyper-parameter tuning that includes FLS?

* Concerning the data model, does it need to satisfy Assumption 4.1? It seems like it would i.h.p. How hard would it be to re-phrase Assumption 4.1 in probability to make it more realistic? Assumption 4.1 seems to be devised to get a stationary point in the Phase 1 ODE. Why not state that as your key assumption?

* Is there any standard benchmark on which Assumption 4.1 may hold? Assumption 4.1 requires the conic hall of all positively labeled datapoints to have angle $\le\pi$. This means that positively labeled data is clustered in a very geometrically- (rather conformally-) specific way. It’s no wonder that this would flesh out a strong alignment component, no?

* Both Fig. 2 and 3 seem to suggest that the FLS sweet spot lies more or less at the same $(c,\eta/c)$ spot, though the manifold of Test Acc around it may differ considerably. (1) What suggests such stability across architectures and intrinsic dimensions? (2) Can we utilize this observation in practice, namely, reduce internal dimension and tune hyper-params for optimal FLS and then use the sweet spot we found for the full fledged task?

* In the data model, I did not understand how the $\mathbf{s}$'s are chosen per data point. Also, the same STD is used for both classes. How difficult would it be to introduce $\sigma_+$ and $\sigma_-$ to make things a bit more interesting?

**Limitations:**

Considering the single paragraph dedicated to limitations and future work at the end, I am concerned that the authors have not fully considered the limitations of their theoretical setup both in its ability to be generalized and in its ability to capture phenomena observed in practice.

* Theory is worked out for a two-layer ReLU NNs trained with logistic loss. Can it be extended or did we hit a theory wall? I suspect the difficulty lies in working out the sub-gradients.

* This work does not consider regression. Is it that the key observation does not apply? In other words, the authors do not justify their focus strictly on classification problems.

* The authors analyze classification problems using logistic loss which is not smooth. This requires using gradient flow with sub-gradients. I could not find it stated in Sec. 3. whether this setup is used in the experiments. If yes, how do experiment results compare with SOTA, in other words, does this training protocol actually interesting for practical purposes? If no, this weakens the value of the analytical toy model.

* The data model used in the analysis is very restrictive in a way that may affect the classification task towards a favorable separation of alignment and over-fitting.

* The experiments use deep convolutional multiclass models, including residual networks. This does not invalidate the theory as a source of intuition, but it weakens any claim that the experiments confirm the proposed mechanism unless the authors clarify which aspects of the mechanism are expected to survive these architectural changes.

**Strengths And Weaknesses:**

Significance:
==========

This paper addresses an important problem in deep learning and tackles it both with sound empirical work as well as with a bold step towards a theoretical explanation; I consider it bold primarily in its attempt to handle finite networks in the rich learning regime rather than at the infinite width limit or assuming lazy learning. Nevertheless, my main concern is that the toy model used in the theoretical analysis is too limited both to be generalized as well as to capture phenomena observed in practice.

Presentation:
==========

The main problem addressed in the manuscript is well stated and well situated within its relevant literature. I enjoyed reading the Introduction and Related Work. The work is also well positioned: from observation of experimental phenomenon to theoretical setting where it can be theoretically established.

Possible typos / adjustment suggestions:
* Notation: define $\mathbb{R}_+$
* Fig 3.: add % to gap bar?
* Phase 1 (L202,Col2): should be “weights are aligned to a particular direction”?
* (L214,Col2): double “is”
* $W_{\max}$ def (L258,Col1): $W_j$ should be $\mathbf{w}_j$?
* Footnote 1: should be “all arguments in this paper hold for any exponentially-tailed loss function”?
* (L262,Col2): should be “of the given model by which cone they belong to.”

Soundness:
=========

I am somewhat concerned about the move from multiclass in experiments to binary in analysis. In the binary theoretical model, prediction is defined only up to a global label flip, so sub-50% accuracy may reflect sign reversal rather than failure. Therefore low accuracy is not, on its own, evidence in favor or against your proposed mechanism.

I am not sure about what’s stated in L266-268 Col1. The notation suggests that $\mathbf{W}(0)$ has been scaled. Thus it would seem that $v_j(0)$ is sampled from a _scaled_ Unif distribution. In that case, even at $t=0$, we aren’t just pulling out an $\alpha$ scalar. So I don’t see how initialization scale factor is essentially equivalent to an output multiplier. I looked at Prop 5.1. It refers to an $L$-layer bias-free, positively homogeneous network without defining what that is. Should I assume $\mathbf{W}_2$ are the $v_j$’s? In that case, the sampling protocol would be as in (1) and not as in (5) for $\mathbf{W}_2$, i.e., the $v_j$’s. Re-defining the sampling protocol for the $v_j$’s may be necessary.

---

> ### Author Rebuttal · Authors · 2026-03-31
>
> Dear reviewer GqSR,
>
> Thank you for your insightful comments.
>
> ---
>
> **W1. Limitations of toy model**
>
> On one hand, it is true that a two-layer ReLU net may not fully capture the complexity of modern networks. On the other hand, the model provides a stylized yet insightful simplification which captures nonlinear feature learning arising in deeper models. This is why the setup (or even simpler linear nets) remain popular for recent works in deep learning theory [1,2].
>
> Moreover, we emphasize that our analysis on this simple model is well-aligned with the empirical phenomena that arise in deep nets (sec. 3).
>
> ---
>
> **W2. Typos and adjustments**
>
> Thank you. We will incorporate them into the revised version.
>
> ---
>
> **W3. Multiclass to binary in analysis**
>
> We sincerely disagree to this point. Our setup considers the case where we achieve low training loss w.r.t. GT label, i.e., no label flip. If the model achieves <50% accuracy on the test data, it should also be deemed a generalization failure.
>
> In fact, using binary classification tasks as a proxy of general classification for theoretical analyses is rather common in the literature (e.g., [3]).
>
> ---
>
> **W4. Weight sampling process**
>
> **Weight sampling procedure.** In short, both $\mathbf{W}, \mathbf{v}$ are scaled up by $\alpha$: $\mathbf{W}=\alpha\mathsf{W}$ and each $v_j(0)$ is sampled so that its magnitude is proportional to $||\mathbf{w}_j(0)||$.
>
> ReLU satisfies $g(cx)=cg(x)$, i.e., positive homogeneity. Thus, scaling up both layers weights by $\alpha$ is the same as scaling the function by $\alpha^2$. Moreover, up to proportional lr scaling, gradient updates remain identical to the output-scaled net as well.
>
> **On Prop B.1.** The reviewer is correct; for Prop B.1, we consider scaling up independently sampled weights of all layers by $\alpha$. In the revised version, we’ll clarify these points.
>
> ---
>
> **Q1. Training time**
>
> Luckily, the cost is not significant. Below, we report #epochs needed to achieve the peak test accuracy on CIFAR-100+ResNet18 (avg over 3 seeds).
>
> ||#epochs|Peak acc|
> |-|-|-|
> |$c=2^0$|$16.33$|$55.03$|
> |$c^\star=2^{-4}$|$18.67$|$59.88$|
>
> The fact that large FLS requires more GD steps is already known, but the cost is as small as ~2 epochs for 4%p accuracy gain, which cannot be achieved by training longer with $c=2^0$.
>
> ---
>
> **Q2,3. About Assm 4.1**
>
> As the reviewer noted, our key assumption is that the data admit a per-class stationary point of the GF ODE; Assm. 4.1 is one sufficient condition.
>
> Moreover, we can ensure that Assm. 4.1 holds w.h.p.; see our response to Q1 of the reviewer xv7o. Also, this assumption is approximately satisfied on image benchmarks, e.g., MNIST [4].
>
> ---
>
> **Q4. Transferring FLS**
>
> As we noted in sec. 6, we believe this is a promising direction.
>
> In simple models, we observe that $c$ is transferable across $n,h$; see our response to Q1 of the reviewer uWMC. In larger models, the transfer across different $n$ seems possible; see our response of S1 to the reviewer xv7o.
>
> ---
>
> **Q5. About data model**
>
> **On $\mathbf{s}$.** Thank you for pointing this out. We assign each datum to $+,-$ uniformly at random (and the per-class mean $\mathbf{s}_{-,+}$ can be chosen arbitrarily).
>
> **With different $\sigma$.** Our framework naturally extends to this setup. The error bound will be an average of the Thm 5.6. Hence the optimal $\alpha$ will be determined by the relative magnitudes of the $\sigma$s. We will add this general form in the final version.
>
> ---
>
> **Limitations**
>
> Following the suggestion, we will substantially expand the limitations section.
>
> - **Can the theory be extended?** The challenge will be to characterize the alignment for deeper models, which needs capturing inter-layer interactions. The sub-gradient analysis seems relatively simpler [4].
> - **Focus on classification.** Although the alignment has also been observed in regression [5], our key tools are specific to the analysis of classifiers. We, however, do not view this as a critical limitation, as it is rather typical in theory works to focus on either classification or regression.
> - **On GF.** We clarify that we used SGD for Sec 3. On one hand, this is a common trick. On the other hand, we believe that this demonstrates the strength of our theoretical framework---the predicted behavior surviving slight shifts in setup.
> - **Restricted data model.** This is a fair point. Relaxing the data model assumptions is indeed an important future work.
> - **Mechanisms surviving architectural changes.** As the “neural collapse” is known to take place in deep nets [6], we suspect that the key mechanisms of OA may survive the architecture change.
>
> ---
>
> [1] Montanari & Urbani, Dynamical Decoupling…, NeurIPS’25
>
> [2] Hashimoto, Directional Convergence…, ICLR’26
>
> [3] Lyu & Li, Gradient Descent Maximizes…, ICLR’20
>
> [4] Min et al., Early neuron alignment…, ICLR’24
>
> [5] Atanasov et al., Neural Networks as Kernel…, ICLR’22
>
> [6] Papyan et al., Prevalence of neural collapse…, PNAS’20

---

> > ### Author Rebuttal · Reviewer_GqSR · 2026-04-02
> >
> > The authors have addressed my concerns satisfactorily. My score remains 4.

---

> > > ### Author Response · Authors · 2026-04-03
> > >
> > > Dear reviewer GqSR,
> > >
> > > We are glad to hear that your concerns have been addressed satisfactorily. If you have no further questions, we would be grateful if you would consider raising your score.
> > >
> > > Best regards,
> > > Authors

---

### Official Review · Reviewer_xv7o · 2026-03-12

**Soundness:** 3
**Presentation:** 3
**Significance:** 4
**Originality:** 3
**Overall Recommendation:** 4
**Confidence:** 3

**Summary:**

This paper studies through a combination of theoretical analysis and experiments how feature learning strength (FLS) influences the generalization error in supervised learning. They demonstrate that in a variety of circumstances there is an optimal intermediate FLS. In computer vision problems across many architectures, they show that an optimal FLS exists. Further, they analyze a two layer network learning a classifier on Gaussian mixtures where they can explain the intermediate optimal FLS as a competition between an over-alignment and over-fitting effect. They argue that over-alignment increases with FLS while overfitting decreases with FLS leading to an optimum.

**Compliance With Llm Reviewing Policy:**

Affirmed.

**Final Justification:**

The rebuttal addressed my main concerns and also promised to add additional high dimensional analysis in the upcoming version. I am now at a weak accept (4).

**Key Questions For Authors:**

1. **Orthogonal separability assumption** The authors provide an othogonality condition on the data in Assumption 4.1. This seems like it could be a strong condition on the empirical distribution that rules out, for instance, any pair of same-class points that are nearly orthogonal. The authors acknowledge this but do not quantify how $λ$ scales with $d$ and $n$ under the Gaussian mixture model. In fact, under the data model of Eq. (6), one can show that λ is O( d^{-1/2} ) with high probability meaning it shrinks in high dimensions. Does this imply that any $\lambda$ dependence in the derived bounds is hiding a $d$ dependence? If so, this is another

2. **An unclear argument about $g(\alpha)$** After Lemma 5.4, the authors write "we may approximately bound $g(α) ≲ O(1)$" The epistemic status of the argument here was unclear to me upon reading, yet this seems to be an important claim to establish equation 22. Can the authors discuss the challenges in establishing the estimate on $g(\alpha)$ and what would be needed to rigorously analyse it?

3. **Stopping Time** The experiments often stop training once a training criterion is reached. Do the results change if one does optimal early stopping for the validation loss (which could occur at a time different than the time taken to reach a train loss threshold)?

**Limitations:**

The authors mention orthogonal separability assumption as a key limitation. I think an additional limitation is not providing a full understanding of how finite data $n$ (or finite $n/d$ in a joint limit) causes the overalignment issue for random covariates in high dimension.

**Strengths And Weaknesses:**

***Strengths***

**Important Question Addressed on FLS**

Many works have been examining the lazy and rich operating regimes of deep networks, but demonstrating that *generalization* can be optimized by an intermediate FLS is an important takeaway. I appreciate this work as clarifying this phenomenon and providing an over-alignment picture for this failure mode of ultra-rich training.

**Quality Experiments**

The experiments clearly disentangle the effect of learning rate and the FLS by conducting joint two-dimensional sweeps. This is a good experimental paradigm to make comparisons across FLS. I would also appreciate some experiments varying the **dataset size** to see the influence on optimal $\alpha$.

***Weaknesses***

**No Proportional Asymptotics**

The over-alignment term OA(α) depends critically on $\phi =  \angle(x₊, s₊)$, the angle between the empirical class mean and the true signal. The paper handles this only in the separate extremes $n\to\infty$ and $d\to\infty$, but the practically relevant regime where both diverge proportionally, say $d/n → γ \in [0,\infty)$, is entirely absent. It would significantly enhance the paper if the optimal FLS $\alpha^\star$ could be computed as a function of $\gamma$.

This could be analyzed for the Gaussian mixture model using the DMFT approach of this [work](https://arxiv.org/abs/2002.11544).


**Clarifying how FLS suboptimality relates to dataset size**

The over-alignment phenomenon is attributed to the predictor aligning too strongly to the empirical class mean rather than to the true signal $s₊$. This is a finite-sample effect — with enough data, $x₊/‖x₊‖ → s₊$. But the paper does not clearly distinguish between over-alignment as a finite-sample bias (the empirical mean is a bad proxy for s₊ in high dimensions) versus as a geometric constraint (the cone $H(α)$ excludes $s₊$ when $α$ is too small). The proportional asymptotics requested above would clarify exactly which effect dominates and under what conditions.

**Bound likely loose**

The over-fitting bound of Theorem 5.6 uses a Rademacher complexity argument (Theorem D.6, Maurer & Pontil 2021) that yields $O( n^{-1/2})$ rates. While sufficient for proving the existence of an optimal α, this bound is likely quite loose — it does not account for the geometry of the cone $H(α)$, and in particular does not exploit the fact that $H(α)$ becomes a narrower cone as $α$ decreases. A tighter analysis would bound the Gaussian complexity of $H(α)$ in terms of $\gamma$.


**Strawman Characterization of Prior Theory Work**

The authors claim in the introduction that "feature learning improves generalization." I am not sure that the consensus is that feature learning always enhances generalization. Many prior theoretical works do not make any claims and rather simply aim to characterize the trajectory resulting over varying feature learning strength such as this [work](https://iopscience.iop.org/article/10.1088/1742-5468/ad01b0/meta). In fact, several works on [continual learning](https://arxiv.org/abs/2506.16884) or [transfer learning](https://arxiv.org/abs/2507.04448) have demonstrated that too high FLS during pretraining can lead to poor performance on downstream tasks.

---

> ### Author Rebuttal · Authors · 2026-03-31
>
> Dear reviewer xv7o,
>
> Thank you for your insightful comments.
>
> ---
>
> **S1. Varying dataset size**
>
> We have evaluated the accuracies of the ResNet18+BigGAN setup, varying the training dataset scale.
>
> |Train set size|x$1.0$|x$.5$|x$.2$|x$.1$|
> |-|-|-|-|-|
> |$c^\star$|$2^{-6}$|$2^{-5}$|$2^{-3}$|$2^{-2}$|
>
> This is close to what our theory predicts, where we derived  $c^\star \propto n^{-2}$. We will add this result in the final version.
>
> ---
>
> **W1. Proportional asymptotics**
>
> First, we clarify that our primary focus is on **moderate** $d$ and $n$. Asymptotic analyses (app. E) are meant to be supplementary, for further intuition.
>
> Nevertheless, we carefully studied [1] as you suggested, and here’s what we think:
>  - It seems challenging to directly extend the ideas of [1] to our setup. Our work analyzes learning dynamics of depth-2 nets under varying FLS with finite-time GF, whereas [1] studies a linear model with explicit regularizer, with no GD/GF involved.
> - We can also extend our result to the joint limiting regime: as $\tilde{\gamma}:=\frac{d}{n^2 \log n}$ (the proportion changed due to Assm 4.1) grows from $0$ to $\infty$, $\phi$ ranges from $0$ to $\pi/2$, and $\alpha^\star$ scales as $\tilde{\gamma}^{1/2}$, in line with the reviewer’s insight.
>
> ---
>
> **W2. FLS suboptimality w.r.t. dataset size**
>
> In the original version, our analysis puts the quantity OA at the center, as it admits a clean closed-form expression via Gaussian CDF (eq. 26).
>
> However, as the reviewer noted, one may write $\mathbf{v}_\star=\mathbf{v}_g(\alpha)+\mathbf{v}_f(n,d)$ to decouple the “geometric constraint” and the “finite-sample bias.” Even so, an exact decomposition is not possible because of the nonlinear property of the Gaussian CDF.
>
> Regarding proportional asymptotics, with $\tilde{\gamma}$ defined in W1:
> - Geometric constraint dominates when $\tilde{\gamma} \to 0$: we have $\mathsf{OA} \to 0$.
> - Finite-sample bias dominates when $\tilde{\gamma} \to \infty$: we have $\mathsf{OA} \to \Phi(0) - \Phi(-1/\sigma)$.
>
> For the moderate $n,d$ we consider, these results do not give the full picture as $n,d$ and $\alpha$ are independent variables: In this case, the OA term is affected by two distinct effects (Eq. (30)).
>
> We will add these results to the final version.
>
> ---
>
> **W3. Bound likely loose**
>
> Thanks for the insightful suggestion. Following this comment, we worked out a geometry-based bound, which is ***complementary*** to our original bound. Specifically, via chaining arguments, we can modify $g(\alpha)$ as $\tilde g(\alpha)/\sqrt{n}\leq C\frac{\sqrt{1-\Psi^2}}{\Psi} \frac{\sqrt{1+\sigma^2}\sqrt{d-1}}{\sqrt{n}}$, for some $C>0$.
>
> While this bound has the same rate as Thm 5.6, it is advantageous whenever $\Psi \to 1$ where we get $\tilde{g}(\alpha)\to 0$. On the other hand, the bound diverges when $\Psi \to 0$ while Thm 5.6. remains finite.
>
> We’ll add these results to the final version.
>
> ---
>
> **W4. Strawman characterization of prior work**
>
> In the standard in-distribution classification (our setup), the prevailing view seems to be that feature learning improves generalization, e.g. [2,3,4]. The reference [5], which reviewer mentioned, also aligns with this view; e.g., Fig.1 (left) of [5].
>
> On the other hand, as the reviewer noted, it seems true that prior works noted that for “broader” classification tasks this may not necessarily be the case. We will down-tone our argument accordingly.
>
> ---
>
> **Q1. Dependency of orthogonal separability**
>
> As the reviewer pointed out, orthogonal separability does exhibit dependence on $d$ and $n$. However, in our setup, we can ensure that orthogonal separability holds w.h.p., e.g., $\kappa^2=\Omega(\sigma^2\sqrt{d\log n})$. Note that this does not change the theoretical result itself. We will add this in the final version.
>
> ---
>
> **Q2. An argument about $g(\alpha)$**
>
> To exactly derive $g(\alpha)$, we need a sharp bound on $\hat{L}(t)$, but such bounds can be derived only in simpler cases (e.g., [6]), and are not available for standard FC nets due to coupled layer dynamics.
>
> To validate this, we provide simulation in App. E.2, supporting our argument.
>
> ---
>
> **Q3. Early stopping with validation loss**
>
> Thank you for this suggestion.
>
> We conducted additional experiments, and observed that **the tendency remains the same,** which suggest that the emergence of the optimal FLS is robust to the choice of stopping criterion. Below is our results on ResNet18+BigGAN:
>
> |$c$|$2^{-10}$|$2^{-8}$|$2^{-6}$|$2^{-4}$|$2^{-2}$|$2^0$|$2^2$|
> |-|-|-|-|-|-|-|-|
> |Peak acc. (80 epochs)|72.92|75.35|**76.62**|75.18|69.22|59.95|49.78|
> |ES w/ val. loss|67.63|70.63|**73.35**|71.53|66.68|56.83|47.20|
>
> ---
>
> [1] Mignacco et al., The role of regularization…, ICML’20
>
> [2] Chizat et al., On Lazy training…, NeurIPS’19
>
> [3] Woodworth et al., Kernel and Rich…, COLT’20
>
> [4] Mehta et al., Extreme Memorization…, ICLR’21
>
> [5] Graldi et al., The Importance of Being…, ICML’25
>
> [6] Moroshko et al., Implicit Bias in Deep Linear…, NeurIPS’20

---

> > ### Author Rebuttal · Reviewer_xv7o · 2026-04-04
> >
> > I appreciate the authors response. I do still think that exact high D asymptotics would improve the paper so I will make my score 4.

---

> > > ### Author Response · Authors · 2026-04-05
> > >
> > > Dear Reviewer xv7o,
> > >
> > > We are very pleased to hear that your concerns have been addressed.
> > > We will incorporate these results, including further theoretical arguments on proportional asymptotics, into the final version of the manuscript.
> > >
> > > Best regards,
> > > Authors

---

### Decision · Program_Chairs · 2026-04-30

**Decision:**

Accept (regular)

**Comment:**

This paper investigates how feature learning strength (FLS) affects generalization in neural networks, presenting both empirical evidence across multiple architectures (VGG, ResNet) and a theoretical analysis for two layer ReLU networks on Gaussian mixtures. All three reviewers converge on a score of 4 (weak accept), recognizing the importance of the question and the clean intuition behind the over alignment vs. over fitting trade off. The empirical methodology, joint sweeps over output multiplier and learning rate, is well designed, and the finding that an intermediate FLS optimizes generalization is practically relevant. The main concerns raised include the gap between the simplified theoretical setting (binary classification, two layer nets, orthogonal separability) and the deep multiclass architectures used in experiments, the looseness of the Rademacher based bounds, and the lack of proportional high dimensional asymptotics. The authors provided substantive rebuttals addressing most concerns, including additional experiments on dataset size, early stopping, and width scaling, as well as a refined chaining based bound, and all reviewers acknowledged their concerns as resolved. While the theoretical contributions remain limited to a stylized setting and the bounds are not tight, the paper offers a valuable conceptual framework (over alignment vs. over fitting) that aligns well with empirical observations and provides actionable guidance for hyperparameter tuning. I recommend acceptance.